# Repair of noise-induced damage to stereocilia F-actin cores is facilitated by XIRP2 and its novel mechanosensor domain

Elizabeth L Wagner[1,2], Jun-Sub Im[1], Stefano Sala[3], Maura I Nakahata[1], Terence E Imbery[1,4], Sihan Li[1,2], Daniel Chen[1], Katherine Nimchuk[1], Yael Noy[5], David W Archer[1], Wenhao Xu[6], George Hashisaki[4], Karen B Avraham[5], Patrick W Oakes[3], Jung-Bum Shin[1,2,4,7]*

[1]Department of Neuroscience, University of Virginia, Charlottesville, United States; [2]Department of Biochemistry & Molecular Genetics, University of Virginia, Charlottesville, United States; [3]Department of Cell & Molecular Physiology, Stritch School of Medicine, Loyola University Chicago, Chicago, United States; [4]Department of Otolaryngology-Head & Neck Surgery, University of Virginia, Charlottesville, United States; [5]Department of Human Molecular Genetics and Biochemistry, Faculty of Medicine and Sagol School of Neuroscience, Tel Aviv University, Tel Aviv, Israel; [6]Genetically Engineered Murine Model (GEMM) Core, University of Virginia, Charlottesville, United States; [7]Department of Cell Biology, University of Virginia, Charlottesville, United States

*For correspondence: js2ee@virginia.edu

**Abstract** Prolonged exposure to loud noise has been shown to affect inner ear sensory hair cells in a variety of deleterious manners, including damaging the stereocilia core. The damaged sites can be visualized as 'gaps' in phalloidin staining of F-actin, and the enrichment of monomeric actin at these sites, along with an actin nucleator and crosslinker, suggests that localized remodeling occurs to repair the broken filaments. Herein, we show that gaps in mouse auditory hair cells are largely repaired within 1 week of traumatic noise exposure through the incorporation of newly synthesized actin. We provide evidence that Xin actin binding repeat containing 2 (XIRP2) is required for the repair process and facilitates the enrichment of monomeric γ-actin at gaps. Recruitment of XIRP2 to stereocilia gaps and stress fiber strain sites in fibroblasts is force-dependent, mediated by a novel mechanosensor domain located in the C-terminus of XIRP2. Our study describes a novel process by which hair cells can recover from sublethal hair bundle damage and which may contribute to recovery from temporary hearing threshold shifts and the prevention of age-related hearing loss.

## Editor's evaluation

Hair cells, the sensory receptors of the inner ear, are easily damaged by exposure to noise. This study provides mechanistic insight into the process by which the mechanosensitive stereocilia of the hair cells can recover from damage-induced gaps in their actin core, possibly allowing for the rescue of transient hearing loss after noise exposure. This manuscript will be of considerable interest to the inner ear field as well as readers with a broader interest in actin cytoskeleton dynamics.

## Introduction

Hair cells, the sensory receptors of the inner ear, are exposed to continuous mechanical stimulation from noise and head movement, which can, in some cases, be harmful to the sensitive structures of the cells, including the apical mechanosensitive apparatus known as the hair bundle (*Wagner and Shin, 2019*). The hair bundle, composed of filamentous (F)-actin-based stereocilia arranged in a staircase-like format, is deflected in response to mechanical stimulation, which causes the opening of mechanotransduction (MET) channels at stereocilia tips (*Hudspeth, 2005*; *Fettiplace, 2017*). Intense stimulation, such as from loud noise, can cause damage to the stereocilia F-actin cores, which is visible by transmission electron microscopy (TEM) as disorganization of the paracrystalline structure of negatively stained actin filaments (*Tilney et al., 1982*; *Engström et al., 1983*; *Liberman, 1987*; *Liberman and Dodds, 1987*) or by confocal light microscopy as 'gaps' in phalloidin-labeled F-actin (*Avinash et al., 1993*; *Belyantseva et al., 2009*). This damage is likely to decrease the rigidity of the bundle (*Saunders et al., 1986a*; *Duncan and Saunders, 2000*), which could lead to reduced MET. In order to preserve hair cell function, an active repair mechanism would be needed to repair any damage to stereocilia F-actin. Passive repair through actin treadmilling is unlikely to be sufficient, because, unlike in most F-actin-based structures, F-actin turnover in stereocilia is mostly restricted to a dynamic zone at stereocilia tips (*Zhang et al., 2012*; *Drummond et al., 2015*; *Narayanan et al., 2015*).

Previous studies (*Belyantseva et al., 2009*; *Andrade, 2015*) found that monomeric β- and γ-actin, as well as the actin-associated factors cofilin and espin, are enriched at phalloidin-negative gaps in stereocilia F-actin. The presence of monomeric actin, along with espin, which can crosslink actin filaments, and cofilin, which has actin severing activity and can nucleate F-actin assembly at high concentrations, led the authors to suggest that localized actin remodeling occurs at these sites to repair the damage. It was also shown that inner hair cell (IHC) stereocilia in mice lacking γ-actin develop gaps in the absence of overstimulation. Therefore γ-actin is likely important for the repair of gaps or in the prevention of their formation. However, they did not conclusively show that this damage can be repaired or how repair might occur.

In this study, we demonstrate that the number of noise-induced gaps in murine hair cell stereocilia actin cores decreases to baseline levels within 1 week of exposure and that newly synthesized actin is incorporated into the stereocilia core in this time frame, suggesting that the damage is repaired through localized F-actin remodeling. We further provide evidence for an involvement of Xin actin binding repeat containing 2 (XIRP2) in this repair process. XIRP2 expression is enriched in hair cells where it colocalizes with F-actin-based structures (*Francis et al., 2015*; *Scheffer et al., 2015*). XIRP2 is expressed in two main isoforms: a long isoform that includes a known F-actin binding domain encoded by exon 7, and a short isoform that lacks exon 7, but includes a LIM domain and an uncharacterized C-terminal domain (CTD). The long isoform primarily localizes to the cuticular plate and cell junctions, while the short isoform is enriched in stereocilia (*Francis et al., 2015*). XIRP2 knockout mice have normal hair bundle development, but F-actin core disruption is visible as early as P7 (*Scheffer et al., 2015*) and stereocilia degeneration is detectable by P12, leading to hearing loss by 7 weeks of age (*Francis et al., 2015*). Consistent with a role in stereocilia repair, we find that XIRP2, mediated by its CTD, localizes to stereocilia F-actin gaps. We further show that in fibroblasts, XIRP2's CTD is recruited to laser-induced stress fiber strain sites, reminiscent of a subset of LIM domain proteins that were previously implicated in F-actin repair (*Uemura et al., 2011*; *Smith et al., 2013*; *Sun et al., 2020*). Our study thus provides evidence for a novel stereocilia repair process, in which XIRP2 senses F-actin lesions in a force-dependent manner, and subsequently orchestrates an actin remodeling process to repair the lesion.

## Results

### Exposure to loud noise causes reversible damage to the stereocilia F-actin core

As was previously reported in guinea pigs (*Avinash et al., 1993*; *Belyantseva et al., 2009*), exposure to prolonged loud noise causes damage to the core of sensory hair cell stereocilia (*Figure 1A–B*), visualized as 'gaps' in phalloidin labeling of F-actin (*Figure 1B and C*), which are most easily observed in the tallest row stereocilia of IHCs. In adult mice, gaps are present in a small percentage of IHCs (3.76%±1.506) in mice unexposed to prolonged loud noise, but following 1 hr of 120 dB broadband

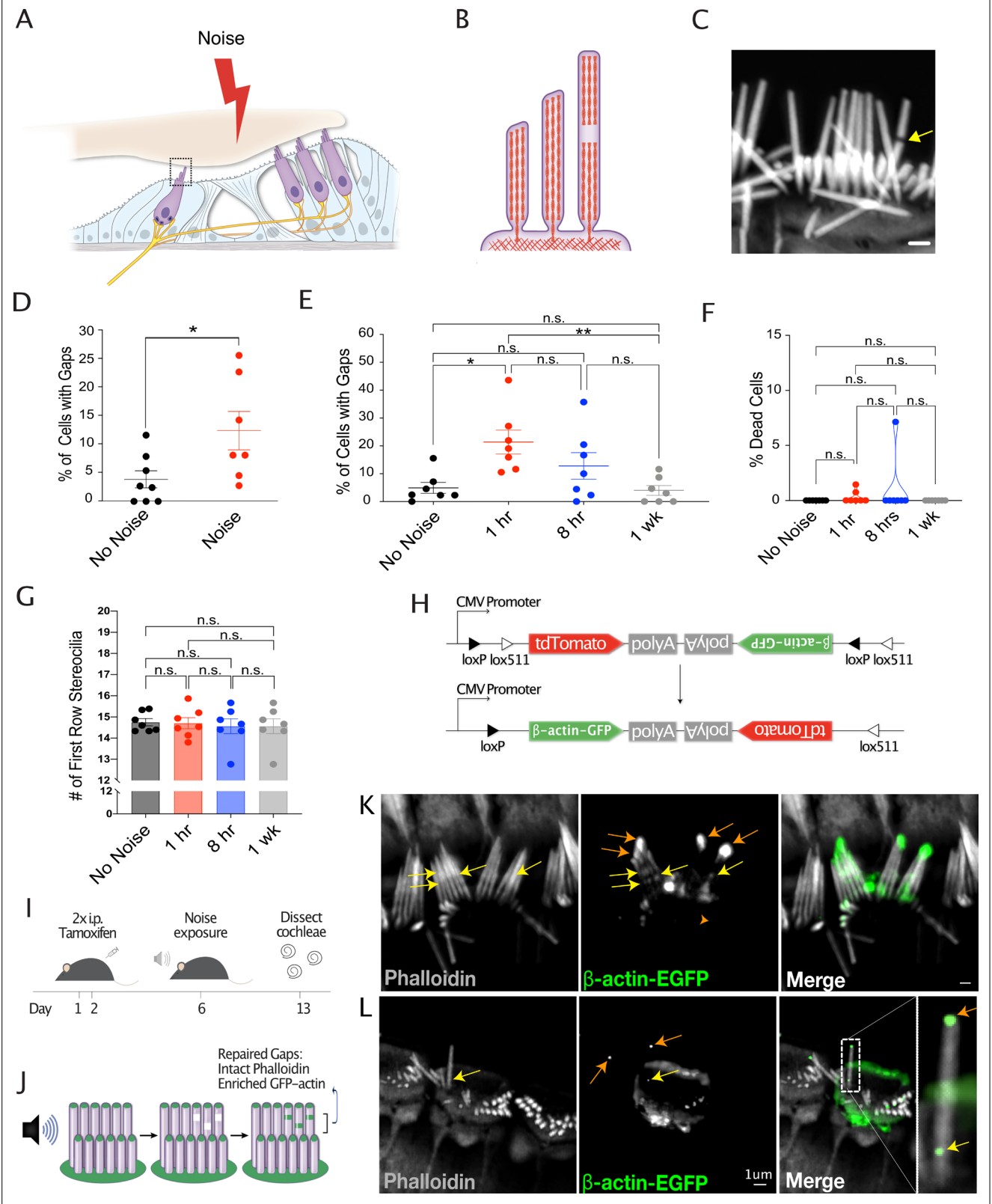

**Figure 1.** Noise-induced lesions in the F-actin cores of stereocilia is repaired by actin remodeling. (**A**) Cartoon showing cross section of the organ of Corti. (**B**) Cartoon depicting side view of an inner hair cell (IHC), with a gap in stereocilia F-actin. (**C**) Representative image of an IHC with a gap in stereocilia F-actin, indicated by arrow. Scale bar: 1 μm. (**D**) Increased percentage of cells with phalloidin-negative gaps in IHC stereocilia F-actin following 1 hr noise exposure (*, p=0.0306). No Noise: n=8 organs of Corti, 4 mice; Noise: n=7 organs of Corti, 4 mice. (**E**) Percentage of cells with

*Figure 1 continued*

gaps initially increases 1 hr following 2 hr noise exposure (p=0.004) but then decreases (p=0.003) to levels not significantly different than in unexposed mice (p=0.74). n=7 organs of Corti, 4 mice per group. (F) Percentage of dead IHCs per cochlea does not significantly change within 1 week of noise exposure. No Noise vs 1 hr - n.s., p=0.974, No Noise vs 8hr - n.s., p=0.521, No Noise vs 1 week - n.s., p>0.999. n=7 organs of Corti, 4 mice per group. (G) Number of tallest row stereocilia per hair cell does not significantly change at any measured point within 1 week of noise exposure. No Noise vs 1 hr - n.s., p>0.999, No Noise vs 8 hr - n.s., p=0.969, No Noise vs 1 week - n.s., p=0.969. (H) Diagram of Cre-mediated inversion in FLEx-β-actin-EGFP mice following tamoxifen injection. Expression of tdTomato is turned off and EGFP-β-actin expression is turned on. (I) Experimental schematic for the observation of the localization of newly synthesized EGFP-tagged β-actin. Mice are injected on days 1 and 2 with tamoxifen and exposed to noise on day 6. Cochleae were dissected and processed on day 13. (J) Cartoon demonstrating the expected localization of EGFP-tagged β-actin in repaired gaps. (K, L) Representative examples from >8 experiments of likely repaired gaps. Yellow arrows point to sites of enriched EGFP-tagged β-actin along stereocilia length with intact phalloidin staining in a Cre-recombined cell following 1 week of recovery from noise exposure. Orange arrows indicate EGFP-labeled stereocilia tips. Due to low recombination rates, the surrounding cells do not express EGFP-β-actin. All error bars represent the standard error of the mean (SEM).

The online version of this article includes the following source data for figure 1:

**Source data 1.** Quantification of stereocilia gap frequency, hair cell death, and number of stereocilia in control and noise-exposed mice.

(1–16 kHz) noise, this increases to 12.31%±3.369 (*, p=0.031) (*Figure 1D*). We chose to focus on IHCs in this part of the study both because noise damages IHC bundles (*Liberman, 1987*) and because of their relative ease of visualization with light microscopy compared to OHCs.

The enrichment of monomeric actin and an actin nucleator and crosslinker at gaps suggested that new F-actin polymerization might be occurring to repair the damage (*Belyantseva et al., 2009*), but direct evidence of this was lacking. In order to determine whether gaps are repaired, we quantified gap frequency (determined as the percent of IHCs with gaps) in a time course following traumatic noise exposure (2 hr, 120 dB broadband noise). Following an initial increase in gap frequency from 4.89±1.97% to 21.39±4.29% in mice 1 hr following noise (*, p=0.013), the percent of IHCs with gaps decreased to 12.80±4.76% by 8 hr post noise (n.s., p=0.312) and to 4.00±1.76% by 1 week post noise (**, p=0.008). Gap frequency 1 week following noise exposure was no longer significantly different from the baseline level (n.s., p=0.998), suggesting that noise-induced gaps are largely repaired within this time frame (*Figure 1E*).

To address the possibility that the decrease in gap frequency was due to the death of damaged cells, we quantified the amount of IHC death at each timepoint before and after noise exposure. There was no significant increase in the number of dead IHCs at any time point after noise (No Noise vs 1 hr, p=0.974, No Noise vs 8 hr, p=0.521, No Noise vs 1 week, p>0.999) (*Figure 1F*), so it is unlikely that the death of damaged hair cells contributed to the observed decrease in cells with gaps. Another potential explanation for the decrease in gap frequency was the shedding of damaged stereocilia. However, we also did not observe any significant change in IHC tallest row stereocilia number following noise exposure (No Noise vs 1 hr, p>0.999, No Noise vs 8 hr, p=0.969, No Noise vs 1 week, p=0.969) (*Figure 1G*), consistent with repair being the primary cause of the decrease in gap frequency.

We hypothesized that gaps repaired by localized remodeling would incorporate newly synthesized actin. Therefore, as an alternative approach to demonstrate the repair of gaps, we examined the localization of actin synthesized following noise exposure. C57BL/6-Tg(CAG-tdTomato,Actb/EGFP)1Erv/J mice (Jackson Laboratories strain #029889, common name FLEx-β-actin-EGFP) were crossed to the tamoxifen-inducible Cre line B6.Cg-*Ndor1*$^{Tg(UBC-cre/ERT2)1Ejb}$/1J (Jackson Laboratories strain # 007001, common name Ubc-Cre$^{ERT2}$), to create the FLEx-β-actin-EGFP+;Ubc-Cre$^{ERT2}$+mice. In this mouse line, *EGFP-β-actin* expression is turned on following Cre recombination (*Figure 1H*; *Narayanan et al., 2015*). Four days following tamoxifen injection, to provide time for maximal Cre recombination, we exposed FLEx-β-actin-EGFP+;Ubc-Cre$^{ERT2}$+mice to traumatic noise (2 hr, 120 dB broadband) and allowed the mice to recover for 1 week (*Figure 1I*), during which time gaps were repaired in our previous experiment. Due to minimal actin turnover in stereocilia, when unexposed to noise, EGFP-β-actin is localized only to stereocilia tips in these mice as previously reported (*Narayanan et al., 2015*; *Figure 1K–L*, red arrows). However, if newly synthesized β-actin is incorporated into gaps during repair, we would expect to see patches of EGFP-β-actin at discrete sites along the length of stereocilia with intact phalloidin staining in addition to the staining at the stereocilia tips (*Figure 1J*). Consistent with this, we see areas of enrichment of newly synthesized β-actin in stereocilia following noise exposure which are likely sites of repair (*Figure 1K–L*, yellow arrows). Due to low Cre recombination rates

(~10% of IHCs) and dim EGFP-β-actin fluorescence, quantification of the occurrence of these 'repaired gaps' was not possible, but their presence supports the complementary evidence that noise-induced stereocilia core damage is repaired.

## XIRP2 is enriched at gaps and is required for their repair

Monomeric β- and γ-actin and several actin-associated factors have been shown to be enriched at gaps, where they likely contribute to the repair process (*Belyantseva et al., 2009*). In addition to these, we observe the enrichment of XIRP2 immunostaining at gaps (*Figure 2A–B*). This enrichment is observed in both naturally occurring gaps in murine auditory IHCs of the cochlea (*Figure 2A*) and vestibular hair cells of the utricle (*Figure 2B*), including gaps which appear to span across several neighboring stereocilia (*Figure 2B*). These adjacent gaps may occur as a consequence of the interconnectedness of the hair bundle, with a mechanical break in one stereocilium destabilizing its neighbors in a cascading manner.

A representative line scan plotting the fluorescence of XIRP2 shows an approximately 1.4-fold increase in XIRP2 intensity in the center of a phalloidin-negative gap compared to the surrounding area of the stereocilia core (*Figure 2C–D*). Occasionally, in larger phalloidin-negative gaps, we observe a gap in the XIRP2 staining as well, with enrichment only at the gap edges (*Figure 2E–F*). XIRP2 enrichment is present both in gaps induced by loud noise exposure (*Figure 3B*) and in gaps occurring in genetic models of hair bundle degeneration (*Goodyear et al., 2012*; *Krey et al., 2018*), including *Ptprq* (*Figure 3C*) and *Elmod1* knockout mice (*Figure 3D*). Additionally, we confirmed that this staining pattern is present in human utricle hair cells, as well (*Figure 3E*). To ensure that the enrichment of XIRP2 immunostaining at these sites was not an artifact of antibody staining, we found that XIRP2 was no longer enriched at gaps in *Xirp2* knockout mice (*Francis et al., 2015*; *Figure 3F*). Moreover, we see XIRP2 enrichment at gaps using a second XIRP2 antibody specific for the short isoform, which is localized to the hair bundle (*Figure 3G*), but not an antibody specific for the long isoform, which is primarily localized to other F-actin-based hair cell structures (*Francis et al., 2015*; *Figure 3H*). A gene map in *Figure 3A* indicates the position of the epitopes for the pan-XIRP2 and isoform-specific antibodies.

Interestingly, XIRP2 appears to remain enriched at repaired gaps, as we see colocalization of XIRP2 enrichment at sites labeled by enriched β-actin synthesized after noise exposure (*Figure 3I*). Consistent with this, we also see an enrichment of XIRP2-enriched sites in stereocilia cores, which does not significantly decrease 1 week following noise exposure (No Noise vs 1 hr, *, p=0.0391; 1 hr vs 8 hr, n.s., p=0.956; 8 hr vs 1 week, n.s., p=0.989; No Noise vs 1 week, n.s., p=0.227) (*Figure 3J*), unlike the number of gaps.

The presence of XIRP2 in gaps suggested that it was playing a role in their repair. To address this possibility, we evaluated the capacity for repair in *Xirp2* knockout mice. We first examined the gaps in vestibular and auditory hair cells that occur in the absence of exposure to loud noise. In *Xirp2* knockout mice, abundant gaps were observed in both vestibular hair cells in the utricle (*Figure 4A*) and in IHCs (*Figure 4C*). At P20, 14.88±1.96% of utricle hair cells *Xirp2* knockout mice had gaps compared to only 4.54±0.70% of cells in wild-type (WT) mice (***, p=0.001) (*Figure 4B*). At the same age, there were also significantly more IHCs with gaps in *Xirp2* knockout mice compared to WT (**, p=0.002) (*Figure 4D*). These results, concordant with a previous report of F-actin filament disorganization in a different *Xirp2*-null mouse line (*Scheffer et al., 2015*), suggest that gaps may build up over time in the absence of XIRP2.

To more directly assess the necessity of XIRP2 in the gap repair, we quantified the percent of IHCs with gaps in a time course following noise exposure (2 hr, 120 dB broadband) in WT and *Xirp2* knockout mice. In WT mice, after an initial increase 1 hr post noise (***, p<0.001), the percent of cells with gaps was no longer significant from baseline levels by 1 week after noise (n.s., p=0.950). However, in *Xirp2* knockout mice, although we observed a smaller, non-significant (p=0.617) initial increase in cells with gaps after noise, the percent of cells with gaps actually further increased from baseline levels during the following week (*, p=0.0140) (*Figure 4E*), rather than decreasing to pre-noise levels like in WT, suggesting that gaps are not efficiently repaired in the absence of XIRP2. We also do not see a significant difference in the number of dead IHCs (*Figure 4F*) or in stereocilia number (*Figure 4G*) in WT or *Xirp2* knockout mice at any timepoint following noise exposure, making it unlikely that hair cell death or stereocilia loss contribute to the observed differences.

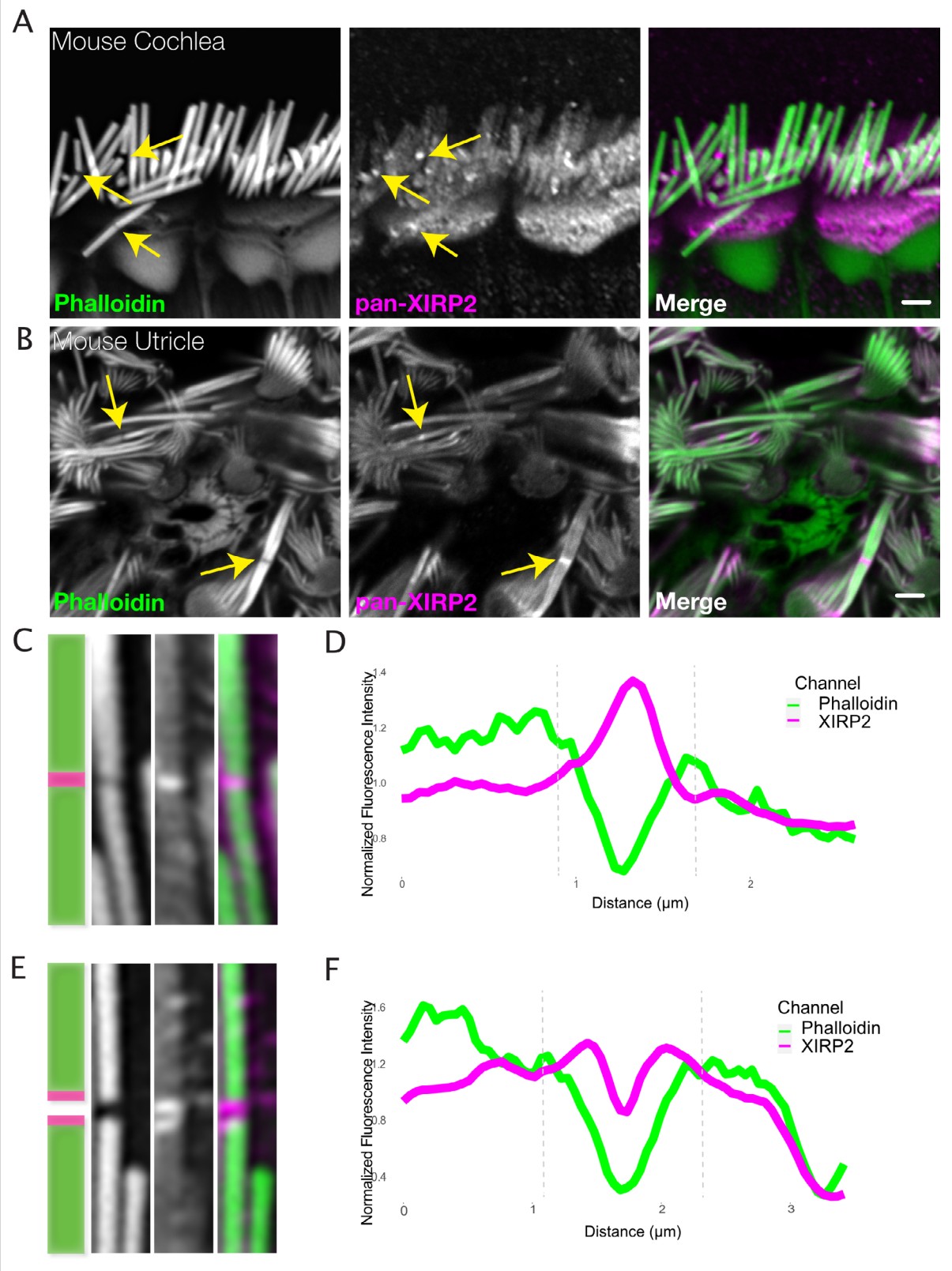

**Figure 2.** Xin actin binding repeat containing 2 (XIRP2) is enriched at gaps in stereocilia F-actin cores. (**A**) XIRP2 immunostaining is enriched at noise-induced gaps in phalloidin staining in inner hair cells (IHCs) (yellow arrows). (**B**) XIRP2 immunostaining is enriched at naturally occurring phalloidin-negative gaps in utricle hair cells (yellow arrows). (**C**) Cartoon and representative image (1×5 µm) of XIRP2 enrichment throughout length of gap in phalloidin signal. (**D**) Line scan of fluorescence intensity in phalloidin (green) and XIRP2 (magenta) channels along the length of a gap in which XIRP2 is

*Figure 2 continued on next page*

*Figure 2 continued*

enriched throughout the gap. (**E**) Cartoon and representative image (1×5 µm) of XIRP2 enrichment at the edges of a gap. (**F**) Line scan of fluorescence intensity in phalloidin (green) and XIRP2 (magenta) channels in which XIRP2 is only enriched at the edges of the gap. Images are representative of >10 experiments.

## XIRP2 is necessary for the enrichment of monomeric γ-actin at gaps

We hypothesized that XIRP2 might be facilitating gap repair through the recruitment of factors necessary for the repair of stereocilia F-actin damage, including monomeric actin and espin. As was previously shown (*Belyantseva et al., 2009*), γ-actin immunostaining is enriched in gaps in hair cell stereocilia. γ-Actin at these sites is likely to be primarily monomeric, as the phalloidin signal is weak, and DNaseI staining, which labels monomeric β- and γ-actin (*Mannherz et al., 1975*), is also enriched at phalloidin-negative gaps (*Belyantseva et al., 2009*; *Figure 5A*).

A representative plot of the fluorescence intensity of γ-actin staining in a utricle hair cell (*Figure 5B and C*) shows an ~2.7-fold increase at the center of the phalloidin-negative gap compared to the surrounding region. However, in contrast to WT mice, γ-actin no longer appears to be enriched in gaps in *Xirp2* knockouts (*Figure 5D and E*). The plot profile in *Figure 5E* shows the absence of γ-actin fluorescence intensity enrichment in an exemplary gap from a *Xirp2* knockout mouse. On average, the ratio of γ-actin fluorescence intensity at the center of phalloidin-negative gaps compared to the edge of gaps decreases from 2.03±0.16-fold in WT utricles to 1.29±0.06-fold in the absence of XIRP2 (***, p=0.0008) (*Figure 5J*). We also found a decrease in the enrichment of γ-actin in noise-induced gaps in *Xirp2* knockout IHCs (**, p=0.0048) (*Figure 5K*).

Although smaller than for γ-actin, we additionally observe a decrease in enrichment of β-actin in gaps from 1.440±0.080-fold in WT (*Figure 5F and L*) to 1.144±0.0314-fold in *Xirp2* knockout mice (**, p=0.0098) (*Figure 5G and L*). Similarly, espin enrichment in gaps is also decreased in *Xirp2* knockouts (***, p=0.0003) (*Figure 5H, I, M*).

## The C-terminal domain of the short isoform of XIRP2 is required for its role at gaps

We reported previously that the sensory hair bundle harbors the short XIRP2 isoform, which is distinguished from the canonical long isoform by the lack of the Xin repeats encoded by exon 7, and by the presence of a LIM domain and a CTD that are encoded by exon 8 and exon 9, respectively. We therefore hypothesized that the short form-specific LIM domain and the CTD might be required for XIRP2's role in the hair bundle, including its recruitment to gaps. To test this possibility, using CRISPR/Cas9-mediated homology-directed repair (*Wang et al., 2013*), we made a mouse in which the LIM domain and CTD are removed. Due to the overlapping but differing reading frames in the C-termini of long and short isoforms of XIRP2, it was impossible to specifically delete the LIM domain and the CTD of short XIRP2 without affecting long XIRP2 (*Figure 6A*). However, by making a 1 bp substitution in exon 8 (T1187A, KM273012.1), we were able to introduce a stop codon before the beginning of the LIM domain in short XIRP2 (L396*, AIR76303.1) without altering the amino acid coding sequence of the long isoform (V3255V, NP_001077388.1) (*Figure 6B*). This mutation led to a truncation of short XIRP2 in which the LIM domain and the CTD are removed (*Figure 6C*). Truncated short XIRP2 (XIRP2-ΔLIM/CTD) does not undergo nonsense-mediated decay, but is still expressed and localizes to the hair bundle, as XIRP2 staining with an antibody recognizing the N-terminus of both isoforms is still present in stereocilia (*Figure 6D*). However, immunostaining with an antibody recognizing the truncated CTD is absent (*Figure 6E*). Confirming that the bundle staining in the *Xirp2$^{ΔLIM/CTD}$* mice was not due to compensatory relocalization of long XIRP2, we found that long XIRP2, recognized by an antibody against the alternative reading frame in exon 8, is still primarily localized to the cuticular plate and cell junctions (*Figure 6F*).

Despite XIRP2-ΔLIM/CTD's colocalization with stereocilia actin cores, accordant with our hypothesis, XIRP2-ΔLIM/CTD shows less enrichment in phalloidin-negative gaps (*Figure 6G*). The enrichment of XIRP2 fluorescence intensity at the center of gaps decreases from 1.27±0.02-fold in WT mice to 0.85±0.03-fold in *Xirp2$^{ΔLIM/CTD}$* mice (***, p<0.0.0001) (*Figure 6H*).

The lack of XIRP2 enrichment in gaps in *Xirp2$^{ΔLIM/CTD}$* mice allowed us to ask whether the localization of XIRP2 to gaps was necessary for the subsequent enrichment of γ-actin. Although truncated short

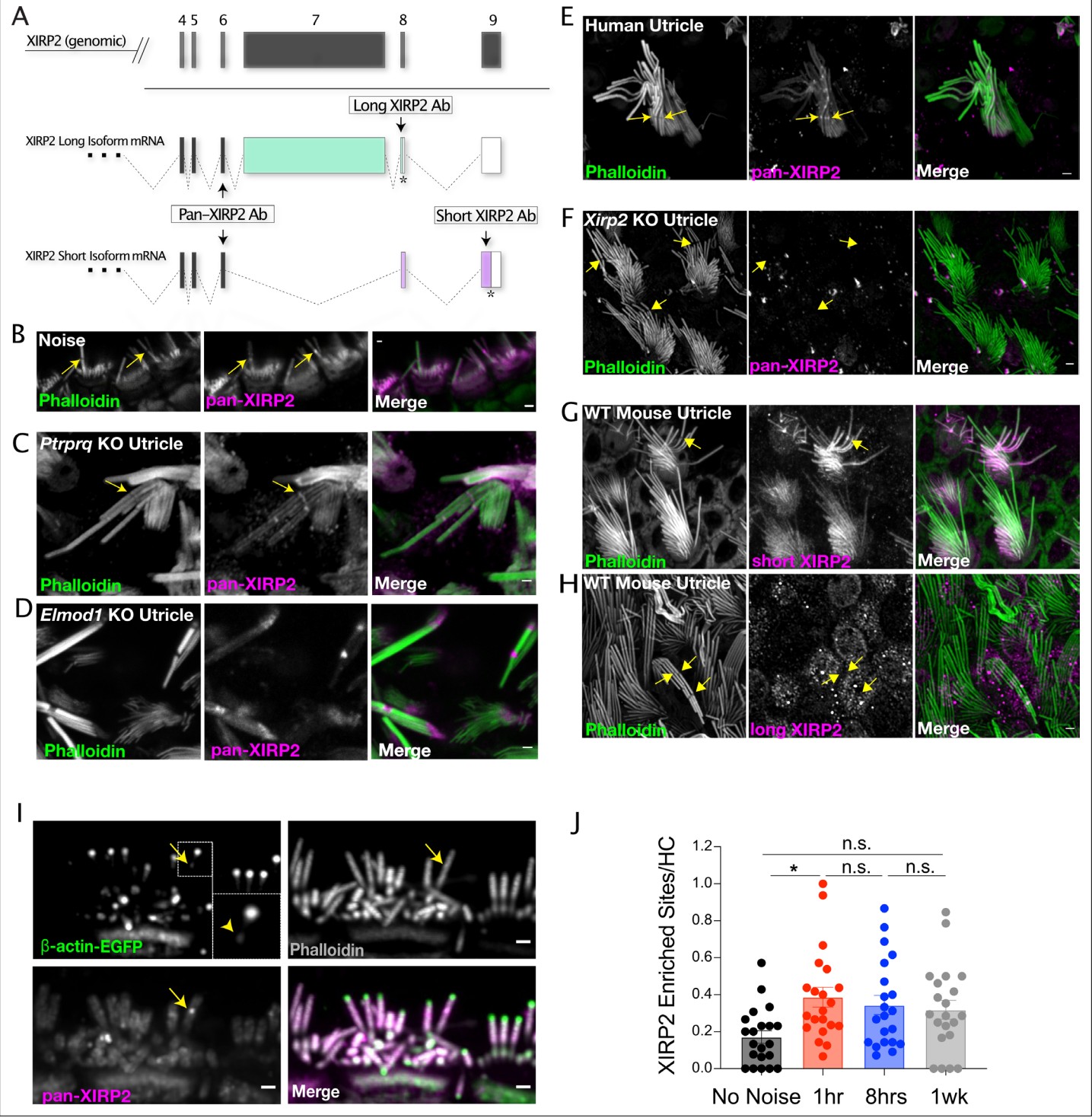

**Figure 3.** The short isoform of Xin actin binding repeat containing 2 (XIRP2) is enriched at gaps and remains there after repair. (**A**) Diagram of *Xirp2* gene structure and isoforms indicating the positions encoding the epitopes recognized by antibodies used in the figure. (**B**) XIRP2 enrichment at gaps (yellow arrows) in inner hair cells (IHCs) induced by overexposure to noise (120 dB broadband [1–22 kHz] for 2 hr). (**C**) XIRP2 enrichment at gaps in a utricle hair cell in a P25 *Ptrpq* knockout mouse. (**D**) XIRP2 enrichment in gaps (yellow arrows) in IHCs in a P20 *Elmod1* knockout mouse. (**E**) XIRP2 enrichment at gaps (yellow arrows) in a human utricle hair cell. (**F**) XIRP2 staining is absent in *Xirp2* knockout utricle hair cells. (**G**) Short XIRP2 is enriched at gaps (yellow arrows). (**H**) Long XIRP2 is excluded from the hair bundle and not present in gaps (yellow arrows). (**I**) XIRP2 enrichment is colocalized with sites of enriched synthesized β-actin synthesized after noise exposure that likely represent repaired gaps (yellow arrow). (**J**) The number of XIRP2 enriched sites in stereocilia increases following noise exposure (No Noise vs 1 hr, *, p=0.039, 1 hr vs 8 hr - n.s., p=0.956, 8 hr vs 1 week - n.s., p=0.979, No

*Figure 3 continued on next page*

*Figure 3 continued*

Noise vs 1 week - n.s., p=0.227). n=7 organs of Corti; 4 mice per group. All scale bars are 1 μm. Error bars represent standard error of the mean (SEM). Images are representative of >3 experiments.

The online version of this article includes the following source data for figure 3:

**Source data 1.** Quantification of number of Xin actin binding repeat containing 2 (XIRP2)-enriched sites in stereocilia in control and noise-exposed mice.

XIRP2 is present in undamaged regions of stereocilia in *Xirp2^ΔLIM/CTD* mice, we observed a decrease in γ-actin enrichment in gaps (*Figure 6I*). The average ratio of γ-actin enrichment is decreased to 1.34±0.09-fold in the *Xirp2^ΔLIM/CTD* mice (***, p<0.001) and is not significantly different from the 1.286±0.057-fold enrichment in WT mice (n.s., p=0.975) (*Figure 6J*), indicating that XIRP2 gap enrichment is necessary for subsequent recruitment of γ-actin.

We next asked whether there are more gaps in *Xirp2^ΔLIM/CTD* mice, as we would predict if XIRP2 and monomeric γ-actin are no longer recruited to gaps to facilitate repair. In line with this, we found an

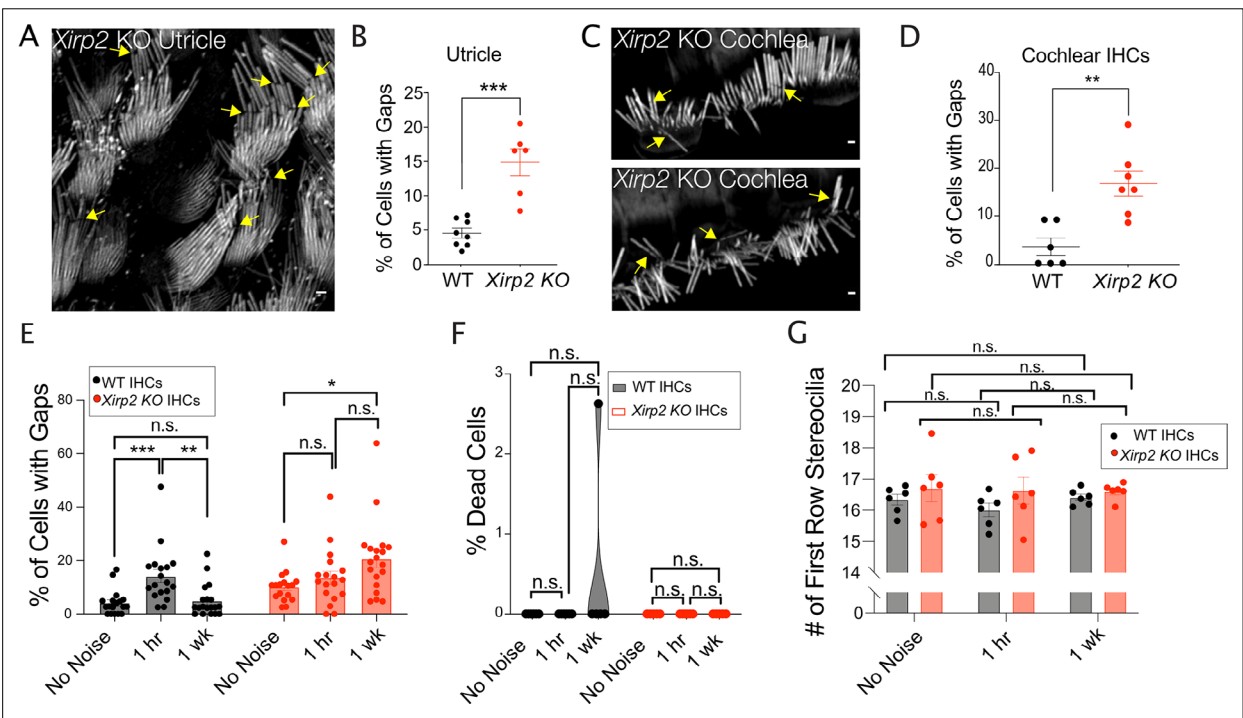

**Figure 4.** Xin actin binding repeat containing 2 (XIRP2) is required for the repair of gaps. (**A**) Gaps (yellow arrows) in stereocilia F-actin in utricle hair cells from P6 *Xirp2* knockout mice. (**B**) There is a significantly larger percentage of utricle hair cells with gaps in P20 *Xirp2* knockout mice than in age-matched wild-type (WT) mice (***, p=0.001). WT: n=8 utricles, 4 mice; *Xirp2* knockout: n=6 utricles, 3 mice. (**C**) Gaps (yellow arrows) in stereocilia F-actin in cochlear inner hair cells (IHCs) from P20 *Xirp2* knockout mice. (**D**) There is a significantly larger percentage of IHCs with gaps in P20 *Xirp2* knockout mice than in age-matched WT mice (**, p=0.002). WT: n=6 organs of Corti; *Xirp2* knockout: n=7 utricles, 4 mice. (**E**) Percentage of cells with gaps decreases to before noise levels within 1 week after initial increase in WT mice, but in *Xirp2* knockout mice, the percentage of cells with gaps further increases during this time period. WT: No Noise vs 1 hr - ***, p<0.001, 1 hr vs 1 week - **, p=0.0013, No Noise vs 1 week - n.s., p=0.9502; *Xirp2* knockout (KO): No Noise vs 1 hr - n.s., p=0.6166, 1 hr vs 1 week - n.s., p=0.1226, No Noise vs 1 week - *, p=0.014. n=18 organs of Corti, 9 mice per group. (**F**) Percentage of dead IHCs per cochlea does not significantly change within 1 week of noise exposure in WT or *Xirp2* knockout mice. WT: No Noise vs 1 hr - n.s., p>0.9999, No Noise vs 1 week - n.s., p=0.3765; *Xirp2* KO: No Noise vs 1 hr - all values 0, No Noise vs 1 week - all values 0. n=6 organs of Corti, 3 mice per group. (**G**) Number of tallest row stereocilia per hair cell does not significantly change within 1 week of noise exposure in WT or *Xirp2* knockout mice. WT: No Noise vs 1 hr - n.s., p=0.3846, No Noise vs 1 week - n.s., p=0.9469; *Xirp2* KO: No Noise vs 1 hr - n.s., p=0.9870, No Noise vs 1 week - n.s., p=0.9769. n=6 organs of Corti, 3 mice per group. All scale bars are 1 μm. Error bars represent standard error of the mean (SEM). Images are representative of >3 experiments.

The online version of this article includes the following source data for figure 4:

**Source data 1.** Quantification of stereocilia gap frequency, hair cell death, and number of stereocilia in wild-type (WT) and *Xirp2 knockout (KO)* hair cells.

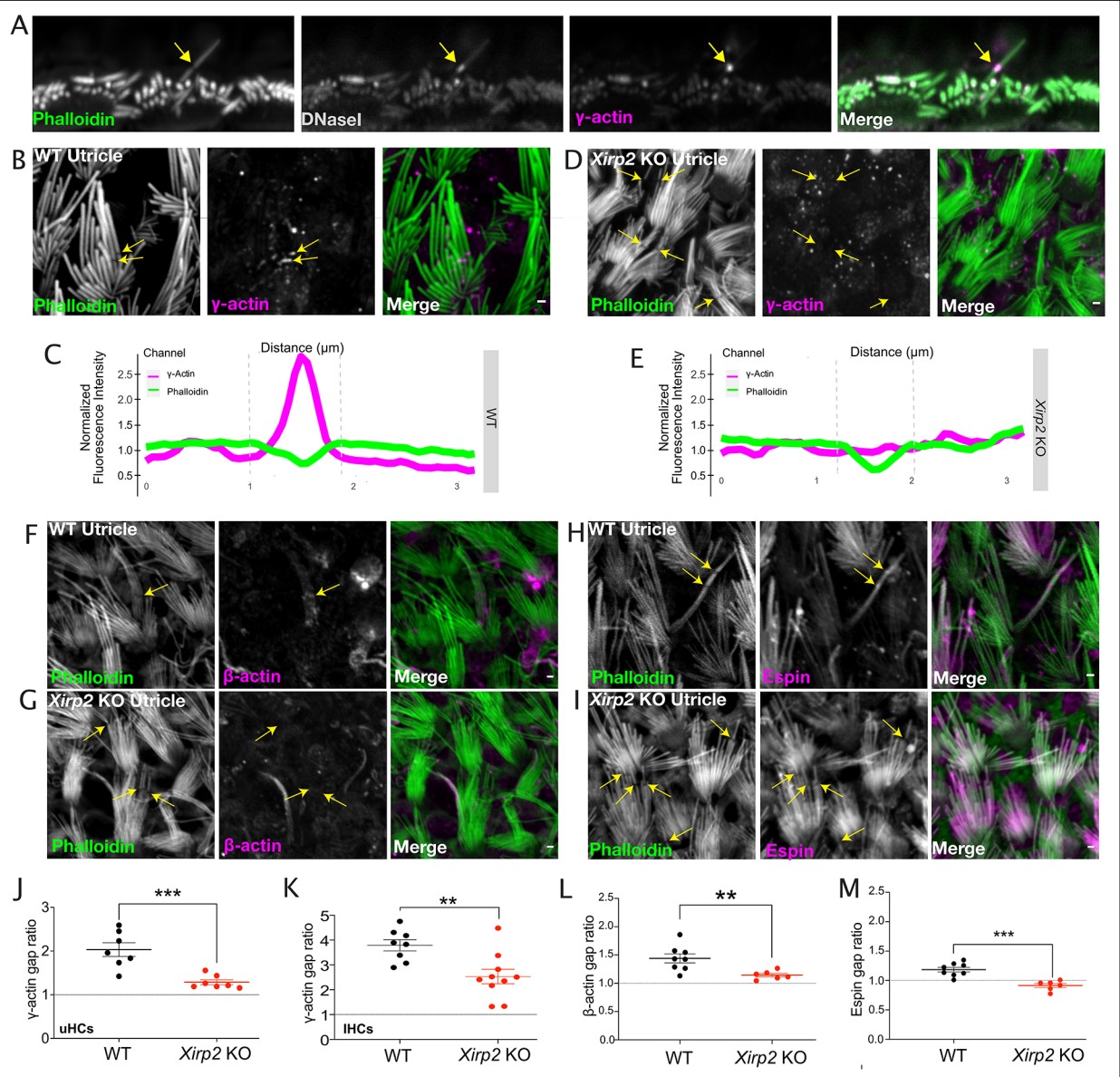

**Figure 5.** Monomeric γ-actin is no longer recruited to gaps in the absence of Xin actin binding repeat containing 2 (XIRP2). (**A**) Representative image of co-labeling of γ-actin and DNaseI enrichment in a noise-induced gap in inner hair cell (IHC) stereocilia F-actin cores. Yellow arrow indicates the site of the gap. (**B**) γ-Actin immunostaining is enriched in gaps (yellow arrows) in utricle hair cells. (**C**) Line scan of fluorescence intensity in phalloidin (green) and γ-actin (magenta) channels along the length of a gap in a wild-type (WT) mouse where γ-actin immunostaining is enriched. (**D**) γ-Actin gap enrichment is decreased in *Xirp2* knockout mice (yellow arrows). (**E**) Line scan of fluorescence intensity in phalloidin (green) and γ-actin (magenta) channels along the length of a gap in a *Xirp2* knockout mouse where γ-actin immunostaining is enriched. (**F**) Representative image of β-actin enrichment at a gap in a WT utricle hair cell. Yellow arrow indicates a gap. (**G**) β-Actin gap enrichment is decreased in gaps in *Xirp2* knockout utricles. Yellow arrow indicates representative gap. (**H**) Representative image of espin enrichment at a gap in a WT utricle hair cell. Yellow arrows indicate gaps. (**I**) Espin gap enrichment in utricle hair cells is reduced in *Xirp2* knockout mice. Yellow arrows indicate gaps. (**J**) The enrichment of γ-actin at gaps, defined as the ratio of the fluorescence intensity at the center of the phalloidin-negative gap to the intensity at the edge of the gap, is decreased in *Xirp2* knockout mice (***, p<0.001). (**K**) The enrichment of γ-actin at noise-induced IHC gaps is decreased in *Xirp2* knockout mice (***, p<0.001). (**L**) The enrichment of β-actin at gaps is decreased in *Xirp2* knockout mice (**, p=0.0098). (**M**) The enrichment of espin at gaps is decreased in *Xirp2* knockout mice (***, p<0.001). All scale bars are 1 μm. Error bars represent standard error of the mean (SEM). Images are representative of >3 experiments.

The online version of this article includes the following source data for figure 5:

**Source data 1.** Quantification of immunofluorescence of γ-actin, β-actin, and espin at stereocilia gaps.

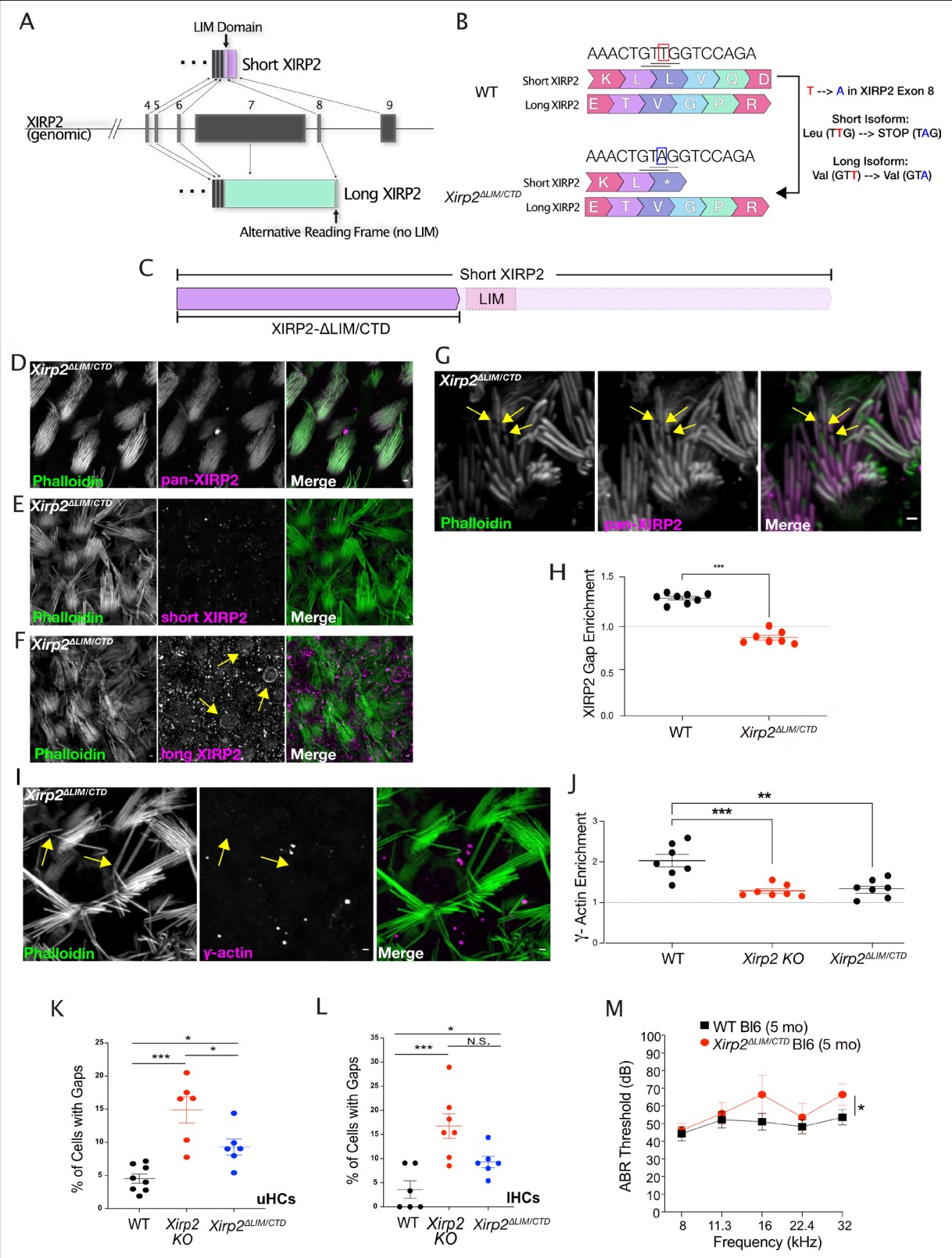

**Figure 6.** The C-terminal domain of Xin actin binding repeat containing 2 (XIRP) is required for its recruitment to gaps, and its deletion in the *Xirp2^ΔLIM/CTD^* mice causes an accumulation of gaps and mild hearing loss. (**A**) Diagram of *Xirp2* gene structure and isoforms indicating the position of the LIM domain and the region of alternative reading frame targeted in the generation of the *Xirp2^ΔLIM/CTD^* mice. (**B**) Diagram indicating the position of the 1 bp substitution in exon 8 of *Xirp2* to generate the *Xirp2^ΔLIM/CTD^* mice. The T→A mutation introduced a stop codon in the short isoform but did not alter the

*Figure 6 continued on next page*

*Figure 6 continued*

amino acid sequence of the long isoform. (**C**) The LIM domain of short XIRP2 and the rest of the C-terminus is removed from short XIRP2 in *Xirp2^ΔLIM/CTD^* mice. (**D**) XIRP2-ΔLIM/C-terminal domain (CTD) (recognized by an antibody against the N-terminus) is still expressed and localizes to the hair bundle. (**E**) Immunostaining with an antibody targeting the CTD of XIRP2 is absent in *Xirp2^ΔLIM/CTD^* mice, indicating the successful truncation of short XIRP2. (**F**) The bundle signal in *Xirp2^ΔLIM/CTD^* mice is not due to compensatory localization of long XIRP2. (**G**) Unlike full-length XIRP2, XIRP2-ΔLIM/CTD immunostaining is not enriched at gaps (yellow arrows). (**H**) Line scan of fluorescence intensity in phalloidin (green) and XIRP2 (magenta) channels along the length of a gap in *Xirp2^ΔLIM/CTD^* mice in which XIRP2 is not enriched. (**H**) The enrichment of XIRP2 immunostaining at gaps is decreased from 1.27-fold in wild-type (WT) mice to 0.85-fold in *Xirp2^ΔLIM/CTD^* mice (\*\*\*, p<0.0.001). n=8 utricles, 4 mice per group. (**I**) γ-Actin immunostaining enrichment is decreased in gaps (yellow arrows) in *Xirp2^ΔLIM/CTD^* mice. (**J**) γ-Actin gap enrichment is decreased from ~2-fold in WT to 1.3-fold in *Xirp2^ΔLIM/CTD^* mice (\*\*\*, p<0.001). n=7 utricles, 4 mice per group. Values for *Xirp2 knockout* mice is same as in *Figure 5J*. (**K**) There is a significantly larger percentage of utricle hair cells with gaps in P20 *Xirp2^ΔLIM/CTD^* mice than in age-matched WT mice type (\*, p=0.0255). WT: n=8 utricles, 4 mice; *Xirp2^ΔLIM/CTD^* mice: n=6 utricles, 3 mice. (**L**) There is a significantly larger percentage of inner hair cells (IHCs) with gaps in P20 *Xirp2^ΔLIM/CTD^* mice than in age-matched WT mice (\*\*, p=0.007). WT: n=6 organs of Corti, 3 mice; *Xirp2^ΔLIM/CTD^* mice: n=7 organs of Corti, 4 mice. (**M**) *Xirp2^ΔLIM/CTD^* C57Bl/6J mice have elevated hearing thresholds compared to WT mice at 5 months of age, as measured by auditory brainstem response (ABR) (\*, p=0.031). n=14 WT mice, 8 *Xirp2^ΔLIM/CTD^* C57Bl/6J mice. All scale bars are 1 µm. Error bars represent standard error of the mean (SEM). Images are representative of >3 experiments.

The online version of this article includes the following source data for figure 6:

**Source data 1.** Quantification of immunofluorescence of Xin actin binding repeat containing 2 (XIRP2), γ-actin at stereocilia gaps, and hair cell numbers and gap frequency in wild-type (WT), *Xirp2 knockout (KO)*, and *Xirp2^ΔLIM/CTD^* mice.

**Source data 2.** Auditory brainstem response (ABR) data of wild-type (WT) and *Xirp2^ΔLIM/CTD^* mice.

increase in the percentage of utricle hair cells (*Figure 6K*) and IHCs (*Figure 6L*) with gaps compared to WT (\*, p=0.0255 and \*\*, p=0.007, respectively).

The *Xirp2^ΔLIM/CTD^* mice also allowed us to ask whether the presence of XIRP2 in gaps is important for hearing function. Albeit not as severe as the phenotype in *Xirp2* knockout mice, *Xirp2^ΔLIM/CTD^* mice exhibit a mild but significant increase in hearing thresholds compared to WT mice, as measured by ABR (\*, p=0.031) (*Figure 6M*), suggesting that gap repair may be important for maintaining hearing sensitivity.

## The CTD of XIRP2 is mechanosensitive

The findings thus far demonstrated that XIRP2 is recruited to stereocilia gaps in a CTD-dependent manner. In a subset of XIRP2-enriched stereocilia gaps we saw a thinning of the phalloidin signal (often near a bent site), presumably representing an area of partial depolymerization of F-actin (*Figure 7A*). Previous work has identified similar structures in stress fibers of adherent cells, such as fibroblasts, which are recognized and repaired by the LIM domain protein zyxin (*Smith et al., 2010*). This repair process relies on zyxin recognizing strained actin filaments via its LIM domains. More recently, a number of other LIM domain proteins have been identified as also being mechanosensitive (*Sun et al., 2020*; *Winkelman et al., 2020*; *Sala and Oakes, 2021*). Because the short isoform of XIRP2 contains a LIM domain, we hypothesized that it was recognizing strained actin in the stereocilia and facilitating repair. To test this hypothesis, we used a focused laser beam to induce strain sites in stress fibers of fibroblasts in the presence of heterologously expressed XIRP2 constructs fused to EGFP and dynamically measured protein recruitment (*Figure 7*; *Smith et al., 2010*; *Sala and Oakes, 2021*).

Expression of full-length XIRP2 partially labeled the actin cytoskeleton (*Figure 7C*), while the single LIM domain of XIRP2 was cytoplasmic (*Figure 7F*). Neither full-length XIRP2 (*Figure 7C–E*) nor the single LIM domain (*Figure 7F–H*), however, were recruited to the induced strain site in the stress fiber (*Videos 1 and 2*). Unexpectedly, however, the isolated CTD displayed a mild localization to the actin cytoskeleton in the basal state (*Figure 7I*) and strong recruitment to the ablation site (*Figure 7I–K*; *Video 3*). The mechanosensitivity of the CTD, but lack of such behavior in the full-length XIRP2 protein, suggests that mechanosensitivity in XIRP2 is regulated, potentially via autoinhibition by the N-terminal domain (NTD). Together these findings establish that the 470 amino acid (aa) long sequence of XIRP2's CTD contains a novel mechanosensor domain, and provide a candidate mechanism for force-dependent recruitment of XIRP2 to stereocilia damage sites.

## XIRP2 interacts with monomeric actin

Once XIRP2 binds to the lesion site, it likely recruits additional repair factors, most importantly monomeric actin that would presumably polymerize to fill the gap. The lack of monomeric β- and γ-actin

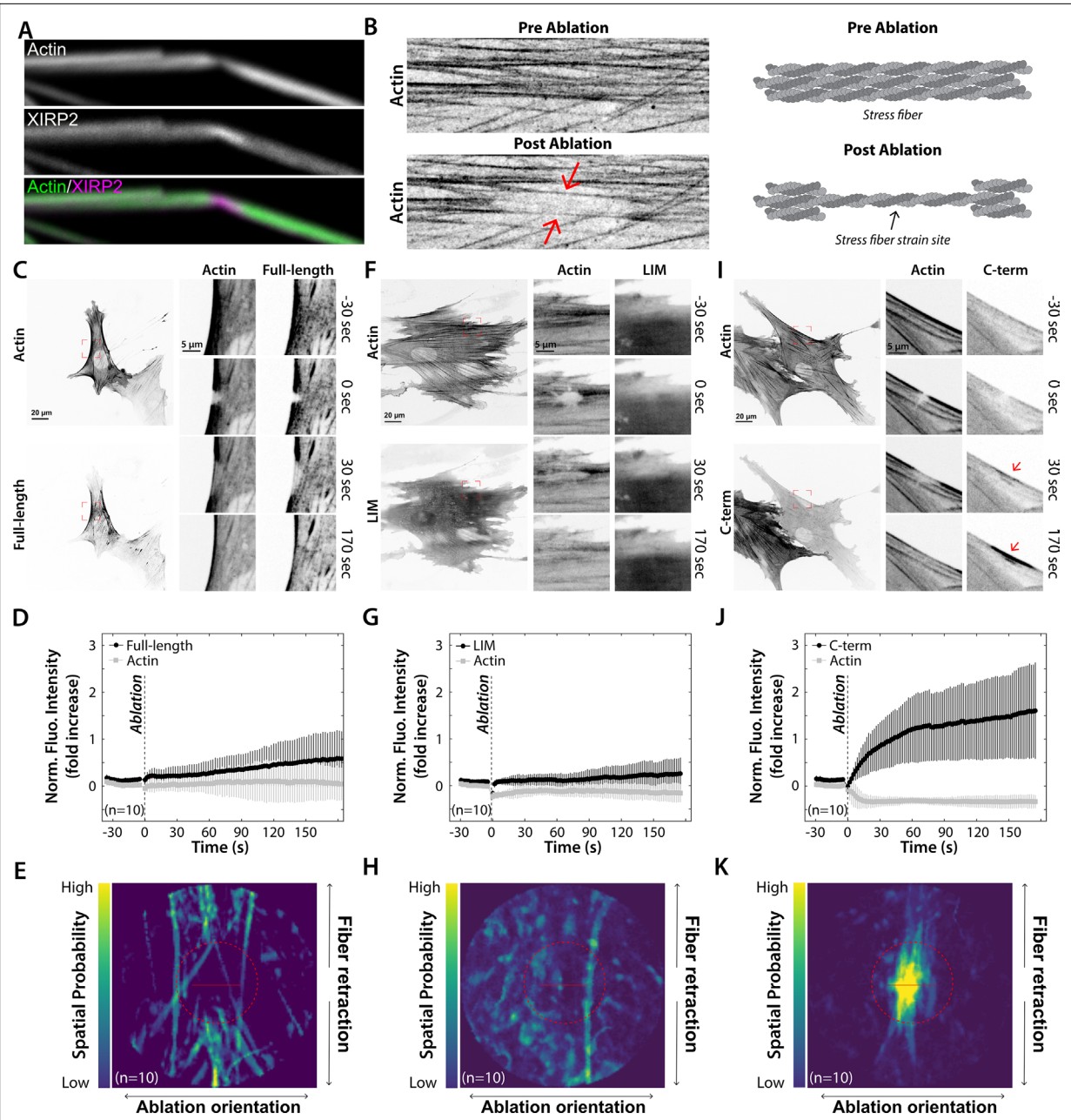

**Figure 7.** The C-terminal domain (CTD) of Xin actin binding repeat containing 2 (XIRP2) recognizes stress fiber strain sites. (**A**) Representative image of a bent stereocilium, characterized by local F-actin depolymerization (reduced phalloidin signal) and enrichment of XIRP2. (**B**) Representative images and cartoon of actin before and after an induced stress fiber strain site. The red arrows indicate thinning actin stress fibers. (**C, F, I**) Human foreskin fibroblasts (HFFs) co-expressing mApple-actin and EGFP-coupled versions of either full length (**C**), the LIM domain (**F**) or CTD (**I**) of XIRP2. The magnified inset images (red boxed regions) show the localization of the actin and XIRP2 constructs during formation of a laser-induced strain site (the ablation timepoint is indicated by '0 s'). The red arrows indicate relocation of the CTD of XIRP2 to stress fiber strain sites (**I**, *Video 3*) which is not observed for the full-length (**C**, *Video 1*) or LIM (**F**, *Video 2*) constructs. (**D, G, J**) Average fluorescence intensity traces and standard deviations of the actin and XIRP2 signals in the region of strain. Following stress fiber strain induction, a large and rapid increase of the fluorescence intensity of the CTD XIRP2 construct is observed (**J**) which is not the case for the full-length (**D**) and LIM domain (**G**) constructs. The slight increase in intensity of full-length XIRP2 during the last minute of the experiment (**D**) is due to the recognition of unstrained actin following stress fiber repair. (**E, H, K**) Spatial probability of XIRP2 recruitment to stress fiber strain sites. The brightest signal of the CTD of XIRP2 is located inside the region of strain (red dashed circle). The brightest LIM domain signal is distributed randomly (**H**) whereas full-length XIRP2 is primarily recruited outside the region of strain indicating it recognizes unstrained actin filaments (**E**). The red line represents the 5 µm ablation line.

The online version of this article includes the following source data for figure 7:

*Figure 7 continued on next page*

*Figure 7 continued*

**Source data 1.** Normalized fluorescence intensity fold changes, of FL (full-length), LIM_only, and C-terminal domain (CTD) (C-term domain) of Xin actin binding repeat containing 2 (XIRP2) tagged with EGFP, after laser ablation.

enrichment at gaps in *Xirp2* knockout mice led us to hypothesize that short XIRP2 can interact directly or indirectly with monomeric actin. This was tested in co-immunoprecipitation assays. Both endogenous γ- and β-actin are co-immunoprecipitated in NIH-3T3 cells transfected with EGFP-tagged short XIRP2, evidence for a direct or indirect binding (*Figure 8A*). When NIH-3T3 cells were co-transfected with EGFP-tagged short XIRP2 and HA-tagged γ-actin or a polymerization-incompetent γ-actin mutant (G13R) (*Posern et al., 2002*), both WT and mutant γ-actin were co-immunoprecipitated (*Figure 8B*), suggesting that XIRP2 has the capacity to bind to monomeric actin. Because γ-actin enrichment was impaired also at gaps in *Xirp2*<sup>ΔLIM/CTD</sup> mice (*Figure 6K*), we hypothesized that the CTD of XIRP2 might harbor the binding site for monomeric actin. Consistent with this hypothesis, both full-length XIRP2 and the CTD were able to co-immunoprecipitate HA-tagged γ-actin (G13R) (*Figure 8C*), suggesting that the CTD is sufficient to bind to monomeric actin.

We also expressed the full length, the NTD, or the CTD of XIRP2 in NIH-3T3 cells and assessed colocalization with phalloidin-stained F-actin. Full length and the NTD displayed a high degree of colocalization with phalloidin, and minimal localization in the cytosol. The CTD, in contrast, was predominantly localized in the cytosol, with occasional localization at a subset of stress fibers (*Figure 8C*).

Taken together, XIRP2 binds, either directly or indirectly, to F-actin, presumably through its NTD, while its interaction with monomeric actin is likely mediated by its CTD. The CTD is thus expected to contain a mechanosensor domain as well as a monomeric actin binding domain.

## *Xirp2* knockout mice are more vulnerable to age-related and noise-induced hearing loss

Our above data demonstrate that stereocilia F-actin gaps can be repaired in a XIRP2-dependent manner. We next asked whether the gap repair is important for the maintenance of hearing sensitivity in aging animals and animals exposed to noise.

We previously reported that *Xirp2* knockout mice, which were maintained on a C57Bl/6J background, develop high-frequency hearing loss by 7–8 weeks of age (*Francis et al., 2015*). Here, we expanded those studies and show that by 11 weeks, the hearing loss worsens to affect all frequencies

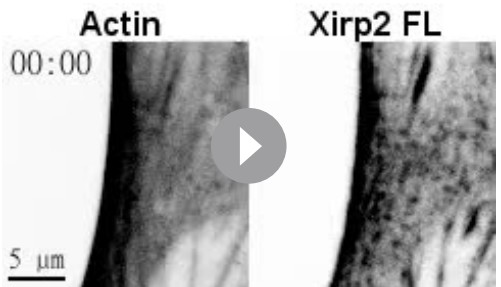

**Video 1.** EGFP-tagged full-length Xin actin binding repeat containing 2 (XIRP2) is not recruited to stress fiber strain sites. A focused laser beam was used to induce strain sites in stress fibers of fibroblasts transfected with mApple-actin and EGFP-tagged full-length XIRP2. Cells were imaged for 3 min and 30 s, alternating between the actin and XIRP2 channel, with images taken approximately every 2 s. After 30 s of imaging the steady state, 5 µm region was illuminated with the 405 laser to induce a strain site via photoablation. The remainder of the time lapse was then imaged. EGFP-tagged full-length XIRP2 showed no recruitment to ablation sites.
https://elifesciences.org/articles/72681/figures#video1

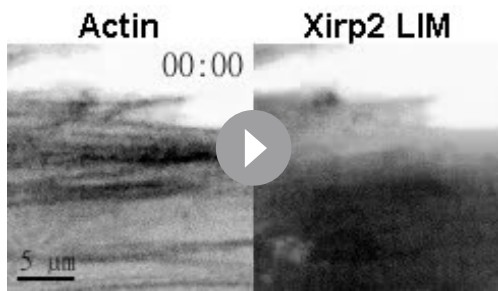

**Video 2.** EGFP-tagged LIM domain of Xin actin binding repeat containing 2 (XIRP2) is not recruited to stress fiber strain sites. A focused laser beam was used to induce strain sites in stress fibers of fibroblasts transfected with mApple-actin and EGFP-tagged XIRP2-LIM domain. Cells were imaged for 3 min and 30 s, alternating between the actin and XIRP2-LIM channel, with images taken approximately every 2 s. After 30 s of imaging the steady state, 5 µm region was illuminated with the 405 laser to induce a strain site via photoablation. The remainder of the time lapse was then imaged. EGFP-tagged XIRP2-LIM showed no recruitment to ablation sites.
https://elifesciences.org/articles/72681/figures#video2

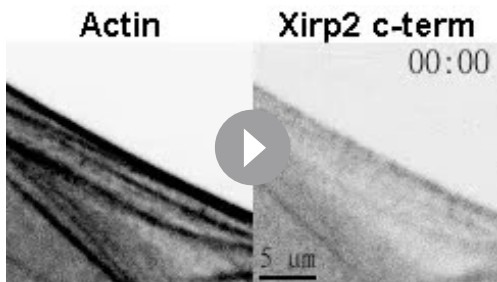

**Video 3.** EGFP-tagged Xin actin binding repeat containing 2 (XIRP2)-C-terminal domain (CTD) shows rapid recruitment to stress fiber strain sites. A focused laser beam was used to induce strain sites in stress fibers of fibroblasts transfected with mApple-actin and EGFP-tagged XIRP2-CTD. Cells were imaged for 3 min and 30 s, alternating between the actin and the XIRP2-CTD channel, with images taken approximately every 2 s. After 30 s of imaging the steady state, 5 μm region was illuminated with the 405 laser to induce a strain site via photoablation. The remainder of the time lapse was then imaged. EGFP-tagged XIRP2-CTD displayed strong recruitment to the ablation site.

https://elifesciences.org/articles/72681/figures#video3

(*Figure 9A*) (***, p<0.001), and progressively worsens over time, as observed at 20 (*Figure 9B*) (***, p<0.001) and 40 weeks (*Figure 9C*) (***, p<0.001). Progressive hearing loss in mice maintained on the C57Bl/6J background is frequently exacerbated due to a mutation in the tip link protein *Cdh23*, which is harbored in this strain (*Johnson et al., 1997*). To separate the effect of this mutation on the worsening hearing in *Xirp2* knockout mice, we also measured hearing function in *Xirp2* knockout mice backcrossed for 10 generations to the CBA/J background. Although the hearing loss is less severe in CBA/J *Xirp2* knockout mice, and ABR thresholds are not significantly elevated at 11–12 weeks of age (*Figure 9D*) (n.s., p=0.090), CBA/J *Xirp2* knockout mice develop a mild hearing loss at all frequencies by 20 weeks (*Figure 9E*) (***, p<0.001), which further progresses by 40 weeks (*Figure 9F*).

We further hypothesized that repair of damaged stereocilia cores could contribute to the observed partial recovery of hearing thresholds following traumatic noise exposure (*Wang et al., 2002*). If this is the case, we would expect to see a similar initial temporary threshold shift (TTS) in WT mice and the gap repair-deficient *Xirp2* knockout mice, but decreased recovery in *Xirp2* knockouts, leading to a larger permanent threshold shift (PTS). To test this possibility, we exposed WT and *Xirp2* knockout mice on CBA/J background to PTS-inducing noise (105 dB for 30 min, 8–16 kHz octave band; *Paquette et al., 2016*), at an age (2 months) where there was no significant baseline hearing loss in CBA/J *Xirp2* knockout mice (*Figure 9G*). Following the noise exposure, we surprisingly saw a smaller TTS in the *Xirp2* knockout mice compared to WT, as indicated by lower hearing thresholds 1 day following noise (**, p=0.007) (*Figure 9H*). However, 2 weeks following the noise exposure, the *Xirp2* knockout mice displayed a larger PTS than WT, with higher hearing thresholds across all measured frequencies at this time point (**, p=0.009) (*Figure 9I*). For each individual mouse, we measured the change in threshold between 1 day and 2 weeks following noise exposure and found a decrease in recovery during this period in *Xirp2* knockout mice (*Figure 9J*) (**, p=0.005), in accordance with our hypothesis that gap repair contributes to recovery from noise-induced hearing loss.

## Discussion

The long-term maintenance of sensory hair cells faces a fundamental challenge: to maximize sensitivity, hair cells are built to be delicate and fragile, yet they have to withstand continuous mechanical stress. A potent capacity for repair must therefore be considered indispensable, especially for mammalian auditory hair cells that are not regenerated. In our study, we provide evidence for a novel process that repairs lesions in the stereocilia F-actin core. We showed that the hair cell protein XIRP2, mediated by its mechanosensitive CTD, senses stereocilia F-actin lesions in a force-dependent manner, and subsequently organizes the repair of the lesion through F-actin remodeling.

Acoustic trauma has been shown to cause disorganization and decreased negative staining of stereocilia actin filaments in TEM imaging (*Tilney et al., 1982*; *Engström et al., 1983*; *Liberman, 1987*; *Liberman and Dodds, 1987*) or gaps in phalloidin staining in confocal imaging (*Avinash et al., 1993*; *Belyantseva et al., 2009*). In agreement with this work, we found an increase in phalloidin-negative gaps following noise, and additionally observed gaps, albeit at a much lower rate, in auditory hair cells and vestibular hair cells in mice that were not exposed to damaging levels of noise, suggesting that the stereocilia cores may also be damaged from continuous exposure to environmental noise or head movement. These changes in actin visualization likely represent areas with decreased actin filament density, where actin filaments have been broken and depolymerized due to mechanical forces from

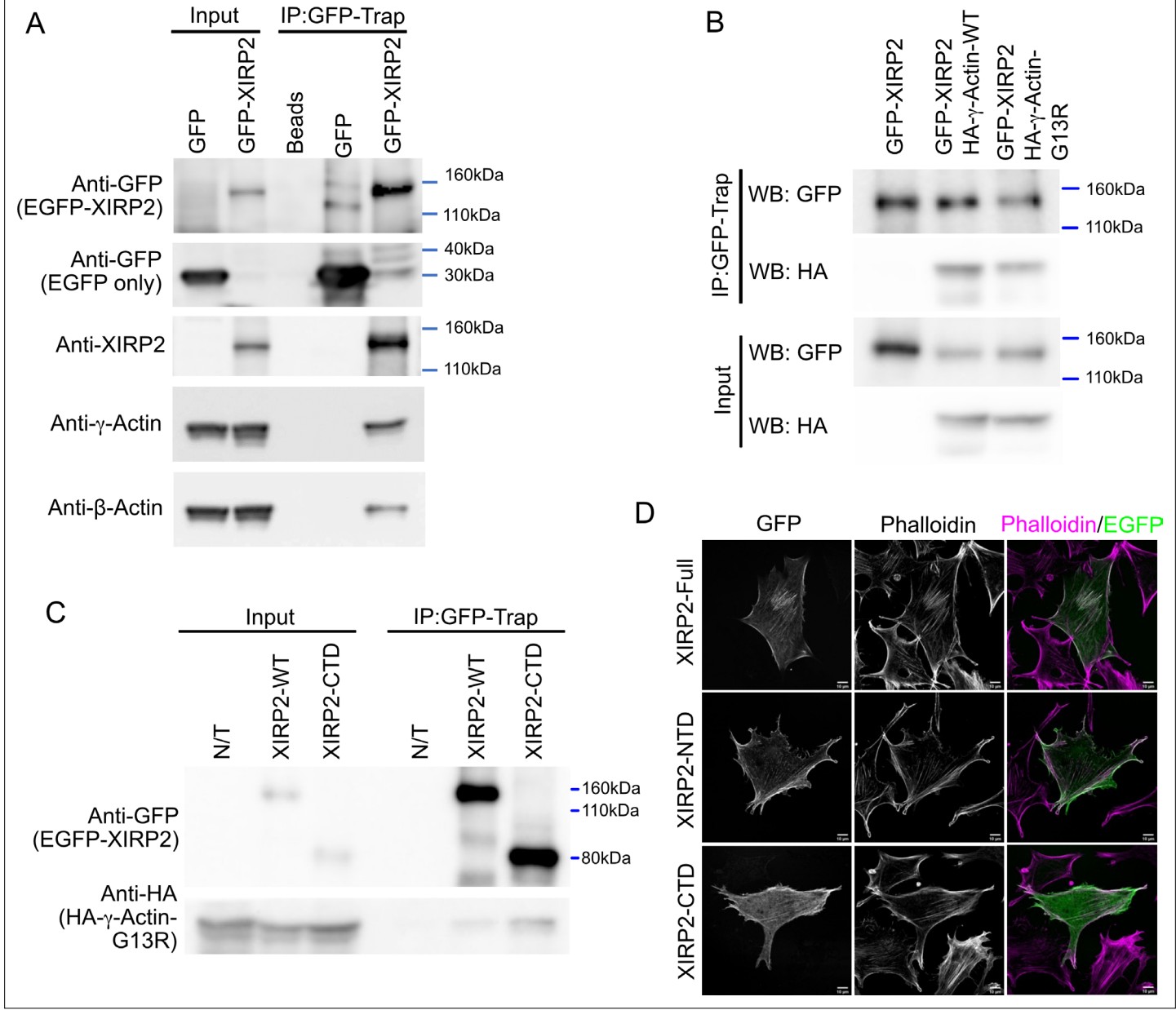

**Figure 8.** Xin actin binding repeat containing 2 (XIRP2) interacts with monomeric and filamentous actin through distinct domains. (**A**) Endogenous β- and γ-actin co-immunoprecipitate with heterologously expressed short XIRP2. NIH 3T3 cells were transfected with the EGFP or EGFP-XIRP2 construct as indicated on the top of each lane. Total cell extract was loaded in input lanes. Immunoprecipitates (IP) were pulled down with GFP-Trap agarose beads followed by western blotting for the indicated antibodies (left). GFP-Trap beads (Beads) were incubated with non-transfected cell extracts as a negative control. (**B**) XIRP2 interacts with monomeric γ-actin. EGFP-XIRP2 WT plasmid was co-transfected with HA-γ-actin wild-type (WT) or the polymerization-incompetent HA-γ-actin G13R mutant. GFP-Trap beads were used for pulldown. Cells transfected with only EGFP-XIRP2 were used as negative control. EGFP-XIRP2 pulls down both the WT and the G13R mutant γ-actin. (**C**) The C-terminal domain (CTD) of XIRP2 is sufficient to pull down monomeric γ-actin. EGFP-XIRP2 WT or the CTD-encoding construct was co-transfected with HA-γ-actin-G13R plasmid. GFP-Trap beads were used for pulldown, and cells transfected with only HA-γ-actin-G13R (N/T) were used as negative control. Both the full-length and the CTD pulldown HA-γ-actin-G13R. (**D**) Full-length XIRP2 and the N-terminal domain (NTD) colocalize with F-actin, while the CTD is predominantly cytosolic and colocalizes only with a select subset of stress fibers. NIH 3T3 cells were transfected with plasmid constructs encoding EGFP-XIRP2 WT, NTD (including the LIM domain) or the CTD, fixed and counterstained with phalloidin.

The online version of this article includes the following source data for figure 8:

**Source data 1.** Western blots showing co-immunoprecipitation (Co-IP) of EGFP-Xin actin binding repeat containing 2 (XIRP2) with β- and γ-actin, Co-IP of EGFP-XIRP2 with wild-type (WT) and polymerization-incompetent γ-actin, and Co-IP of EGFP-tagged full-length and C-terminal domain (CTD) of XIRP2 with polymerization-incompetent γ-actin.

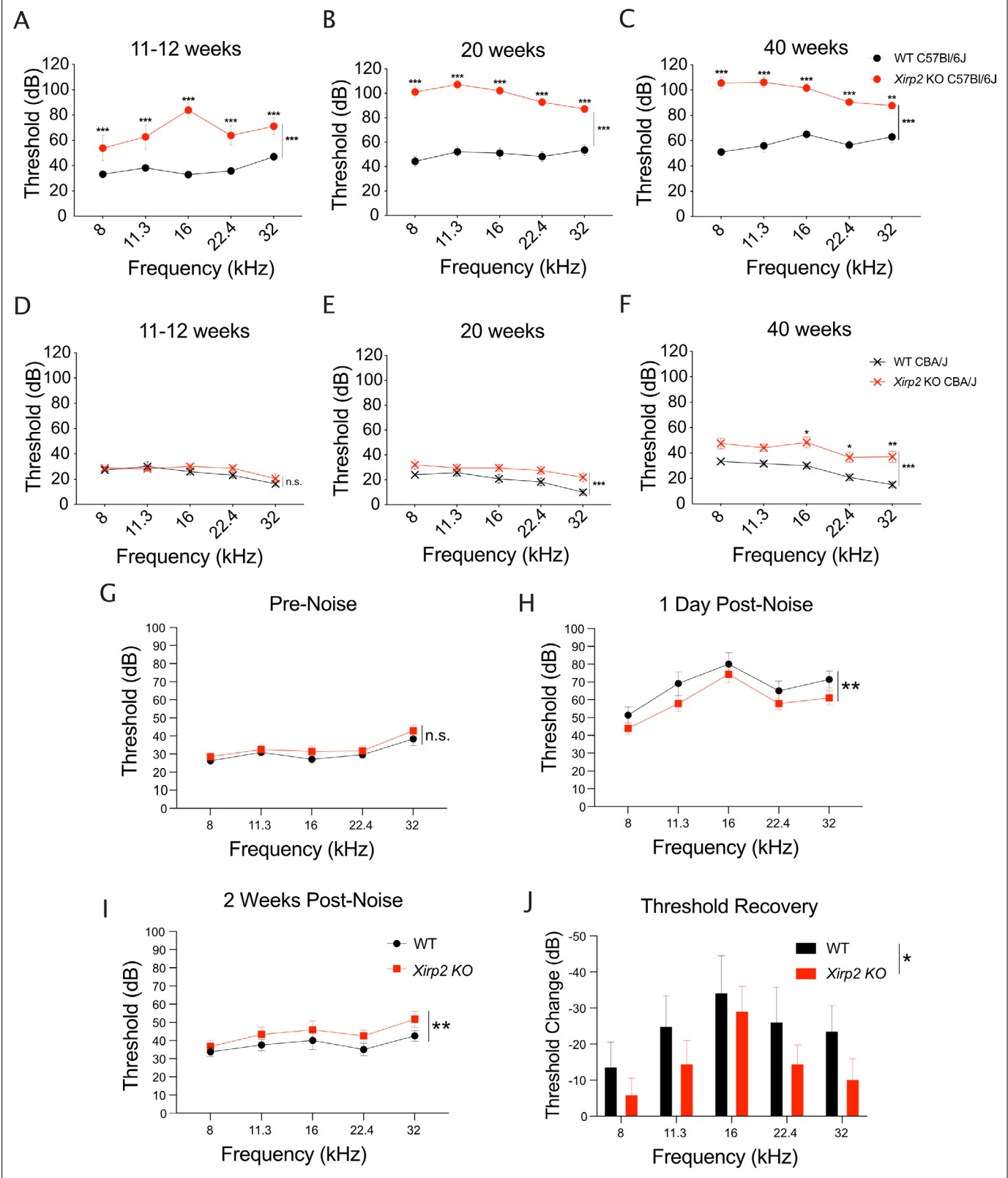

**Figure 9.** *Xirp2* knockout mice are more susceptible to age-related and noise-induced hearing loss. (**A**) Auditory brainstem response (ABR) thresholds are significantly elevated in *Xirp2* knockout (KO) mice on the C57Bl/6J background at 11–12 weeks of age (***, p<0.001). n=17 wild-type (WT), 9 *Xirp2* KO. (**B**) ABR thresholds are further elevated in C57Bl/6J *XIrp2* KO mice at 18 weeks (***, p<0.001). n=14 WT, 9 *Xirp2* KO. (**C**) ABR thresholds are progressively increased in C57Bl/6J *Xirp2* KO mice at 40 weeks (***, p<0.001). n=10 WT, 9 *Xirp2* KO. (**D**) ABR thresholds are not significantly elevated in

*Figure 9 continued on next page*

*Figure 9 continued*

*Xirp2* KO mice on the CBA/J background at 11–12 weeks of age (n.s, p=0.090). n=11 WT, 12 *Xirp2* KO. (**E**) By 18 weeks, ABR thresholds are significantly elevated in CBA/J *Xirp2* KO mice (***, p=<0.001). n=6 WT, 12 *Xirp2* KO. (**F**) ABR thresholds are further increased in CBA/J *Xirp2* KO mice by 40 weeks of age (***, p=<0.001). n=6 WT, 12 *Xirp2* KO. (**G**) Prior to noise exposure, *Xirp2* knockout mice backcrossed to the CBA/J background do not have significantly different ABR hearing thresholds than WT CBA/J mice (n.s., p=0.058). n=12 WT mice, 14 *Xirp2* KO mice. (**H**) Following a noise-induced temporary hearing threshold shift (1 day following noise exposure (105 dB octave band, 1 hr)), WT CBA/J mice have elevated ABR hearing thresholds compared to XIRP2 KO CBA/J mice (**, p=0.007). n=12 WT mice, 14 *Xirp2* KO mice. (**I**) Following 2 weeks recovery, *Xirp2* KO CBA/J mice have elevated ABR hearing thresholds compared to WT CBA/J mice (**, p=0.009). n=8 WT mice, 14 *Xirp2* KO mice. (**J**) *Xirp2* KO CBA/J mice have decreased ABR hearing threshold recovery during the 2 weeks following exposure to noise. (**, p=0.005). n=8 WT mice, 14 *Xirp2* KO mice. All error bars represent standard error of the mean (SEM).

The online version of this article includes the following source data for figure 9:

**Source data 1.** Auditory brainstem response thresholds of aged and noise-exposed *Xirp2* knockout (KO) mice on C57Bl/6J or CBA/J background.

overstimulation or changes in stereocilia ion concentrations from increased MET, as phalloidin binds at the interface of actin monomers within filaments (*Barden et al., 1987*; *Cooper, 1987*). Further studies will be needed to dissect the mechanical versus chemical contributions to the generation of noise-induced lesions in stereocilia cores.

Even in the absence of other hair bundle damage that occurs in concert with the stereocilia breakage during noise exposure (*Wagner and Shin, 2019*), we expect that gaps in the F-actin core decrease the structural integrity of the hair bundle and lead to impaired MET. Previous work has shown that in vitro fluid jet overstimulation of the hair bundle causes a decrease in bundle stiffness (*Saunders et al., 1986a*; *Saunders et al., 1986b*; *Duncan and Saunders, 2000*) without breakage of interstereociliary links (including the tip links), which was proposed to be due to damage to the stereocilia core or to the rootlet where the stereocilia is inserted into the cuticular plate (*Duncan and Saunders, 2000*). Consistent with our observed repair of stereocilia cores through actin remodeling, recovery of bundle stiffness was observed (*Saunders et al., 1986a*; *Duncan and Saunders, 2000*), dependent on hair cell metabolic activity (*Saunders et al., 1986a*). The recovery occurred on a faster time scale than that which we observed, but it is possible that the intensity of damage was less severe due to the in vitro nature of the experiment or that recovery time is variable between species. Additionally, we cannot rule out repair of damage to the stereocilia rootlet, which was difficult to evaluate with confocal microscopy, contributed to stiffness recovery in their experiments. Further investigation of the actin core ultrastructure will be necessary to evaluate this possibility.

The vulnerability of stereocilia cores is compounded by the extremely slow turnover of F-actin. In many actin-based structures actin is continuously turned over by a process known as treadmilling, in which actin monomers are added to the plus-end of the filament at the same rate as they are removed from the minus end, maintaining the filament at a constant length (*dos Remedios et al., 2003*). In contrast, stereocilia actin is stable, except for a small dynamic region at the very tips (*Zhang et al., 2012*; *Drummond et al., 2015*; *Narayanan et al., 2015*), preventing the damaged region from being replaced passively through turnover and necessitating an active repair mechanism to restore bundle rigidity and MET function. Actin monomers, an actin crosslinker, and an actin nucleator, which could serve as building blocks for new F-actin assembly, were found to be enriched at phalloidin-negative gaps (*Belyantseva et al., 2009*), supporting this possibility. Combined with the decrease in gaps we found after recovery from noise exposure and incorporation of β-actin synthesized following noise exposure in discrete patches along the stereocilia shaft, we posit that these factors are recruited to gaps to facilitate the assembly of new actin polymers to replace the damaged filaments. Although we do observe newly synthesized β-actin in repaired gaps, it is possible that depolymerized actin from damaged filaments is also recycled during the repair process, which, together with the low Cre recombination rate, might explain the low frequency at which were able to observe repaired gaps harboring newly synthesized actin and the dim signal at these sites compared to stereocilia tips. As we did not observe EGFP-labeled β-actin along the whole length of the stereocilia following noise, we propose that short filaments spanning the length of the gap are synthesized and attached to existing undamaged regions of the actin core through direct annealing, which has been described in vitro (*Murphy et al., 1988*). Although actin polymerization at the minus end of the filament is less energetically favorable than at the plus end, another possibility is that actin monomers are added on to filaments

on both sides of the gap to fill in the damaged region and then the pieces are annealed together, possibly supported by the split staining of XIRP2 on either side of some gaps.

Actin monomers and the actin crosslinker espin appear to be recruited to gaps in an XIRP2-dependent manner, as demonstrated by the decrease in β-actin, γ-actin, and espin gap enrichment in *Xirp2* knockout mice. We also found that heterologously expressed XIRP2 interacts with both monomeric β- and γ-actin and this interaction is presumably important for their recruitment to gaps. The decrease in monomeric actin gap localization is potentially the primary reason for the lack of gap repair that we observe in *Xirp2* knockout mice, as evidenced by increased gaps in *Xirp2* knockout mice in the absence of noise trauma and lack of decrease in gaps following noise exposure.

We previously reported that among the multiple XIRP2 isoforms expressed in hair cells, short XIRP2 (Genbank ID KM273012) specifically and exclusively localizes to the hair bundle (*Francis et al., 2015*). Immunostaining with isoform-specific antibodies demonstrated that only short XIRP2 is enriched at gaps in phalloidin staining. In contrast to the long XIRP2 isoform, short XIRP2 contains a LIM domain followed by the CTD. Our studies on the *Xirp2^{ΔLIM/CTD}* mice, where the LIM domain and CTD of the short isoform are removed, demonstrated the importance of these domains for targeting XIRP2 to the F-actin lesion sites.

Our efforts to identify the mechanism by which XIRP2 is recruited to the lesion sites were guided by an emerging concept in the field of mechanobiology. It has long been appreciated that extrinsic forces can modify the interaction between F-actin and its binding proteins (*Galkin et al., 2011*; *Jégou and Romet-Lemonne, 2021*). The interaction of F-actin with proteins such as the Arp2/3 complex (*Risca et al., 2012*; *Pandit et al., 2020*), ADF-cofilin (*Pavlov et al., 2007*; *Mizuno et al., 2018*; *Wioland et al., 2019*) and alpha-catenin (*Mei et al., 2020*) is modified ('tuned') by mechanical forces (*Sun and Alushin, 2022*). A subset of LIM domain proteins, however, is unique in that their affinity to F-actin is 'switched on' by binding to strained F-actin (*Sun et al., 2020*; *Winkelman et al., 2020*; *Sala and Oakes, 2021*). These proteins have been previously shown to recruit repair proteins to these strain sites (*Uemura et al., 2011*; *Smith et al., 2013*; *Sun et al., 2020*), leading us to hypothesize that XIRP2 might be recruited to F-actin lesion sites and facilitate repair in a similar manner. While previous studies have suggested that strain recognition requires multiple LIM domains in series (*Sun et al., 2020*; *Winkelman et al., 2020*), the first LIM domain from the protein testin was recently shown to recognize strained F-actin on its own (*Sala and Oakes, 2021*). Surprisingly, however, it was the isolated CTD of XIRP2 that we found was sufficient for recruitment to these strain sites. Interestingly, mechanosensitivity is not found in the full-length XIRP2 protein, suggesting that XIRP2's mechanosensitivity is regulated, potentially in an autoinhibitory manner, reminiscent of the reported behavior of testin (*Sala and Oakes, 2021*).

It is our leading hypothesis that the CTD of XIRP2 constitutes a novel class of mechanosensitive protein domain. Further in vitro studies with purified proteins are needed to demonstrate that the mechanosensitivity of XIRP2's CTD is an intrinsic property, and not mediated by an interacting protein. The CTD consists of ~470 aa, spanning half the length of the short isoform of XIRP2, and lacks any predicted peptide domains. Future studies will determine the exact mechanosensitive domain, and the structural features that mediate force-dependent interaction with F-actin. The CTD also helps recruit additional repair factors including monomeric actin and the crosslinker espin to gaps, suggesting that XIRP2 functions not only as a damage sensor, but also as an organizer of the subsequent actin remodeling efforts.

A crucial question lingers concerning the applicability of force-dependent repair mechanisms observed at stress fibers to stereocilia actin repair. Stress fibers, characterized by their contractile nature and exposure to tensile forces, differ from the F-actin core of stereocilia, where such force conditions have not been reported. Nonetheless, it is plausible that high frequency and amplitude deflection of hair bundles induces localized stress along the shafts of stereocilia, causing focal destabilization and depolymerization of actin. Consequently, this process may cause bending or torsion of the remaining actin fibers, providing a binding signal for XIRP2.

Interestingly, XIRP2 appears to remain at damaged sites after they have been repaired, as we see XIRP2 co-enriched with patches of newly synthesized EGFP-β-actin at presumed sites of repair and the number of XIRP2-enriched sites following noise exposure remains elevated, despite the decrease in gaps during this time period. One possibility for XIRP2's continued presence might be that gaps are not fully repaired over the 1-week period following noise exposure, but this is unlikely unless there

is very minor damage remaining which cannot be observed with the resolution of light microscopy. However, if this is not the case, it may indicate that XIRP2 has a further role following the repair of the actin core, possibly in reenforcing the structure at vulnerable sites.

The *Xirp2^{ΔLIM/CTD}* mouse allowed us to partially separate XIRP2's function in gaps from its other, as of yet uncharacterized, roles in hair cells. Truncated short XIRP2 is still expressed in these mice and still appears to be localized normally in the hair bundle, except it is no longer enriched in gaps. If, as we suspect, XIRP2-ΔLIM/CTD is still able to perform the other functions of XIRP2, the build-up of unrepaired gaps in the *Xirp2^{ΔLIM/CTD}* mouse likely leads to the development of hearing loss in these mice, even in the absence of noise trauma. However, the mild hearing phenotypes suggest that the role of XIRP2 in undamaged regions of the stereocilia core may also play an important role in the maintenance of hearing function.

The connection between repair of gaps and hearing loss is also supported by the progressive increase in hearing thresholds in *Xirp2* knockout mice. The build-up of unrepaired gaps in the absence of XIRP2 may lead to the worsening phenotype, suggesting that gap repair could be necessary for the prevention of age-related hearing loss. Interestingly, the phenotype seems to be exacerbated when the *Xirp2* mutation was backcrossed to the C57Bl/6J strain, which harbors a mutation in the tip-link protein, CDH23. An intriguing possibility is that weakened or non-functional tip links make hair cells more sensitive to the development of gaps and intensify the effect of the loss of XIRP2.

We also propose here that repair of gaps may contribute to partial recovery from noise-induced hearing loss. Repair of other hair bundle structures like tip links (*Zhao et al., 1996*; *Indzhykulian et al., 2013*), which are broken following noise exposure (*Pickles et al., 1987*; *Husbands et al., 1999*), has been hypothesized to contribute to the decrease in hearing thresholds following an initial large TTS. The increased susceptibility of *Xirp2* knockout mice to permanent noise-induced hearing loss is consistent with actin core repair also contributing to this threshold recovery. Although the initial TTS was unexpectedly smaller in *Xirp2* knockout mice compared to WT, the recovery over the 2-week period following noise exposure was dampened in *Xirp2* knockouts, leading to mildly, but significantly, elevated hearing thresholds. The relatively small difference between genotypes was unsurprising, however, if repair of gaps is only one of many factors contributing to this recovery, most of which we do not predict to be altered in the absence of XIRP2.

Despite our functional data demonstrating progressive hearing loss and a reduced ability to withstand noise damage in *Xirp2* knockout mice, it is difficult to directly address the effects of gap formation and repair on hearing function because we do not have a method to specifically induce stereocilia core damage in vivo without damaging other hair cell structures. Much research on age-related and noise-induced hearing loss focuses on hair cell death and damage to the stria vascularis, which maintains the ion concentrations necessary for MET (*Wu et al., 2020*). However, there is evidence to suggest that unrepaired damage to the hair bundle, possibly including actin core lesions, also plays a role in the development of noise-induced hearing loss (*Liberman and Dodds, 1987*). More recently, a human cadaver analysis from the Liberman lab concluded that hearing performance in humans might better correlate with hair bundle morphology than hair cell numbers (*Wu and Liberman, 2022*), further highlighting the significance of hair bundle damage for understanding progressive hearing loss. This study also implied that hair cells might persist with damaged hair bundles in a hypo-functional state, opening up a treatment window for strategies to boost hair bundle repair and function.

This study sheds further light on the importance of repair of sublethal damage to hair cells in order to preserve hearing function following traumatic noise exposure and in aging animals. Due to the inability of mature mammalian hair cells to regenerate (*Groves, 2010*; *Burns and Corwin, 2013*), repair mechanisms are necessary to restore MET and prevent hearing loss. Herein, we add stereocilia actin cores to the growing list of hair cell structures, including tip links (*Indzhykulian et al., 2013*), and controversially, synapses (*Kujawa and Liberman, 2009*; *Shi et al., 2013*), capable of recovery in mammalian hair cells.

## Methods

### Animal care and handling

The care and use of animals for all experiments described conformed to NIH guidelines. Experimental mice were housed with a 12:12 hr light:dark cycle with free access to chow and water, in standard

laboratory cages located in a temperature and humidity-controlled vivarium. The protocols for care and use of animals were approved by the Institutional Animal Care and Use Committees at the University of Virginia, which is accredited by the American Association for the Accreditation of Laboratory Animal Care. C57BL/6J (Bl6, from Jackson Laboratory, ME, USA) and CBA/J (from Jackson Laboratory, ME, USA) mice and sibling mice served as control mice for this study. Neonatal mouse pups (postnatal day 0 [P0]–P5) were sacrificed by rapid decapitation, and mature mice were euthanized by $CO_2$ asphyxiation followed by cervical dislocation. The FLEx-β-actin-EGFP (C57BL/6-Tg(CAG-tdTomato,Actb/EGFP)1Erv/J, Jackson Laboratories strain #029889) mice were purchased from The Jackson Laboratory. The *Ubc-Cre^ERT2* mice (B6.Cg-*Ndor1^{Tg(UBC-cre/ERT2)1Ejb}*/1J, Jackson Laboratories strain #007001) were gifted from Dr. Alban Gaultier (University of Virginia). All procedures involving *Elmod1 knockout* animals met the guidelines described in the National Institutes of Health Guide for the Care and Use of Laboratory Animals and was approved by the Animal Care and Use Committees of Tel Aviv University (M-12-046).

## Generation of *Ptprq* knockout and *Xirp2^{ΔLIM/CTD}* mice

The online tool CRISPOR (http://crispor.tefor.net/crispor.py) was used to select a suitable CRISPR target sequence to knockout *Ptprq* (CTCACCCGTGAGTAGAACAC, in exon 2) and to edit the sequence in XIRP2 to generate the *Xirp2^{ΔLIM/CTD}* mice (GAAACTGTTGGTCCAAGACA, in exon 8 and the preceding intron). The following repair template was used to generate the 1 bp substitution in exon 8 to introduce a stop codon in the short isoform: CCAGTTAGTGTAGTATTTGTTTTGTTAT GCCTTTACAGTTGTGACTGAAAATAGAAATATATAAGCCTTTATATATATTTTTTTTATTTTTAGAAACTGT **A**GGTCCAAGGCAAGGAAATTTGCATAATTTGTCAAAAGACAGTTTATCCAATGGAGTGCCTCATA GCAGACAAGCAGAATTTTCATAAGTCTTGCTTCAGA (1 bp T→A substitution indicated by bolded A). The repair template was ordered as a custom Ultramer DNA oligo from Integrated DNA Technologies. The single-guide (sg)RNA to target exon 8 of *Xirp2* was generated by T7 in vitro transcription of a PCR product from overlapping oligonucleotides (as described in the CRISPOR online tool). Oligonucleotide sequence for the *Xirp2^{ΔLIM/CTD}* knock-in forward primer: GAAATTAATACGACTCACTATAGG AAACTGTTGGTCCAAGACAGTTTTAGAGCTAGAAATAGCAAG, and for the universal reverse primer: AAAAGCACCGACTCGGTGCCACTTTTTCAAGTTGATAACGGACTAGCCTTATTTTAACTTGCTATTTC TAGCTCTAAAAC. The sgRNAs to target exon 2 of *Ptrpq* was generated by in vitro transcription of the target sequence cloned downstream of the T7 promoter in the pX330 vector (Addgene). In vitro transcription was performed using the MAXIscript T7 kit (Life Technologies) and RNA was purified using the MEGAclear kit (Thermo Fisher Scientific). For production of genetically engineered mice, fertilized eggs were injected with precomplexed RNP (ribonucleoprotein)-Cas9 protein (PNA Bio, 50 ng/μl) and sgRNA (30 ng/μl). In the case of the *Xirp2^{ΔLIM/CTD}* mice, the repair template was co-injected (50 ng/μl). Two-cell stage embryos were implanted on the following day into the oviducts of pseudopregnant ICR female mice (Envigo). Genotyping was performed by PCR amplification of the region of interest (*Ptprq* forward primer: ACTTTGGCATTCCAGGTTGATGT, *Ptprq* reverse primer: ATGCAAAGCAAACTCG GCCAAT; *Xirp2-ΔLIM/CTD* forward primer: AGGCTCCCCATAAGTTTGCT, *Xirp2-ΔLIM/CTD* reverse primer: TGCTTGTCTGCTATGAGGCA), followed by Sanger sequencing. Founder mice were selected and backcrossed for seven generations to the C57Bl/6J background. The *Ptprq* knockout mouse harbors a reading frame-shifting 1 bp deletion in exon 2. The *Xirp2^{ΔLIM/CTD}* mice have the intended 1 bp substitution with no other edits.

## Generation of *Elmod1* knockout mice

The ES cells of *Elmod1*-deficient mice (Elmod1^{tm1a (EUCOMM)Hmgu}) were produced at the EUCOMM Consortium and the mice were produced at MRC Harwell. Mice carrying a cassette including LacZ and neomycin resistance genes inserted into intron 4–5 of the *Elmod1* gene, located on chromosome 9. The EMMA ID is EM:04834, common strain name: HEPD0557_4_E10, and international strain designation: C57BL/6N-Elmod1^{tm1a(EUCOMM)Hmgu}/H, are referred to herein as *Elmod1* knockout. The mice were maintained on a C57BL/6NTac genetic background and were bred with WT C57BL/6. For genotyping, genomic DNA was extracted from a small clip of the tail and was used as a template for PCR with specific primers. For the WT allele the primers used were: forward, GGGGTGACCTAAGACCATCA; and reverse, GAGGGCTCGGGAGACATAGT. For the mutant allele: forward, GTTTGTTCCCACGGGAG

AATC; reverse, CCGCCACATATCCTGATCTT. The mutant line is available from the Infrafrontier Mouse Disease Models Research Infrastructure (https://www.infrafrontier.eu/).

## Hearing tests in mice

ABRs of C57Bl/6J WT, CBA/J WT, *Xirp2* knockout C57Bl/6J, *Xirp2* knockout CBA/J, and *Xirp2$^{\Delta LIM/CTD}$* mice were determined at the time point indicated for longitudinal ABRs. Additionally, ABRs of CBA/J WT and *Xirp2* knockout mice were determined at 5–6 weeks on the day prior to and after noise exposure. An additional ABR was performed on the same mice at 2 weeks following noise exposure. Mice were anesthetized with a single intraperitoneal injection of 100 mg/kg ketamine hydrochloride (Fort Dodge Animal Health) and 10 mg/kg xylazine hydrochloride (Lloyd Laboratories). ABRs were performed in a sound-attenuating cubicle (Med-Associates, product number: ENV-022MD-WF), and mice were kept on a Deltaphase isothermal heating pad (Braintree Scientific) to maintain body temperature. ABR recording equipment was purchased from Intelligent Hearing Systems (Miami, Fl). Recordings were captured by subdermal needle electrodes (FE-7; Grass Technologies). The noninverting electrode was placed at the vertex of the midline, the inverting electrode over the mastoid of the right ear, and the ground electrode on the upper thigh. Stimulus tones (pure tones) were presented at a rate of 21.1/s through a high-frequency transducer (Intelligent Hearing Systems). Responses were filtered at 300–3000 Hz and threshold levels were determined from 1024 stimulus presentations at 8, 11.3, 16, 22.4, and 32 kHz. Stimulus intensity was decreased in 5–10 dB steps until a response waveform could no longer be identified. Stimulus intensity was then increased in 5 dB steps until a waveform could again be identified. If a waveform could not be identified at the maximum output of the transducer, a value of 5 dB was added to the maximum output as the threshold.

## Human utricle preparation

Human utricles used in this study were obtained as a byproduct of a surgical labyrinthectomy. Patient information was de-identified. The samples were fixed immediately after removal in a 10% formalin solution.

## Immunofluorescence

Inner ear organs were fixed in 4% paraformaldehyde (Electron Microscopy Sciences) immediately after dissection for 20 min. Samples were washed three times with phosphate-buffered saline (GIBCO PBS, Thermo Fisher Scientific) for 5 min each. After blocking for 2 hr with blocking buffer (1% bovine serum albumin [BSA], 3% normal donkey serum, and 0.2% saponin in PBS), tissues were incubated in blocking buffer containing primary antibody at 4°C overnight. Samples were washed again three times for 5 min with PBS and then incubated in blocking buffer containing secondary antibody at room temperature (RT) for 1.5 hr. Following secondary antibody incubation samples were washed three times for 5 min with PBS and then mounted on glass slides in ProLong Glass Antifade Mountant (P36984, Invitrogen) with a #1 coverslip. The following primary antibodies were used in this study: goat polyclonal pan-XIRP2 antibody used in *Figure 2A–B* (D18 [discontinued], Santa Cruz Biotechnology, Inc, 1:200), rabbit polyclonal pan-XIRP2 antibody used in *Figure 2C–D*, *Figure 3*, and *Figure 6* (custom generated against the immunizing peptide MARYQAAVSRGDTRSFSANVMEESDVCTVPGG LAKMKRQFEKDKMTSTCNAFSEYQYRHESRAEQEAIHSSQEIIRRNEQEVSKGHGTDVFKAEMMSHLE KHTEETKQASQFHQYVQETVIDSPEEEELPKVSTKILKEQFEKSAQENFLRSDKETSPPAKCMKKLLVQDKE ICIICQKTVYPMECLIADKQNFHKSCFRCHHCSSKL, Proteintech, 1:100), mouse monoclonal IgG2a GFP antibody (A11120, Invitrogen, 1:300), rabbit polyclonal long XIRP2 antibody (11896-1-AP, Proteintech, 1:100), rabbit polyclonal short XIRP2 antibody (custom generated against the immunizing peptide NSKRQDNDLRKWGD, Genscript, 1:100), mouse monoclonal IgG$_1$γ-actin antibody (1–24: sc-65635, Santa Cruz Biotechnology, Inc, 1:100). Alexa Fluor-488, -555, and -–647-conjugated donkey anti-mouse IgG, -goat IgG, and -rabbit IgG secondary antibodies (Invitrogen, 1:200) were used to detect appropriately paired primary antibodies. F-actin was detected using Alexa Fluor-488 and -647-conjugated Phalloidin (Invitrogen, 1:200). Fluorescence imaging was performed using an inverted Zeiss LSM880 confocal microscope with a fast-mode Airyscan detector and a 63×/1.4 NA, oil-immersion objective, an inverted Olympus FV1200 confocal microscope equipped with a 60×/1.35 NA oil-immersion objective or an Leica Stellaris 5 confocal microscope equipped with a 63×/1.4 NA oil-immersion objective.

Where indicated, representative images result from maximum intensity projections of several consecutive optical Z-sections.

## Image denoising

To improve the visibility of gaps and enrichment of immunostaining at these sites in representative images, denoising of confocal images was performed using the Noise2Void package, which is based on deep learning. The detailed mathematical model and network architecture for Noise2Void was previously published by the Jug group (*Krull et al., 2020*) and the package can be downloaded at https://github.com/juglab/n2v, (*Krull, 2023* copy archived at swh:1:rev:84f2de54c187237ef91db5e-19fae30548e4ce88f). An Intel i9-10900KF CPU and a single NVIDIA RTX3080 GPU were used for transfer learning. For each of the raw images, we trained a unique Noise2Void convolutional neural network model. In the transfer learning process, raw images of 1000×1000 pixels in x-y dimension were split into multiple 48×48 pixel tiles to make a training dataset. For training of each model, the network was trained with 200 epochs and 100 steps in each epoch. Finally, using the raw image as input, a denoised image can be predicted by using the trained neural network.

## Induction of Cre recombination

To induce the translocation of Cre to the nucleus in FLEx β-actin-EGFP+;Ubc-Cre$^{ERT2}$+mice, mice were intraperitoneally injected with tamoxifen. Two injections were performed 24 hr apart. Tamoxifen (T5648, Sigma) was dissolved at 5 mg/ml in corn oil (C8267, Sigma) at 37°C for 2 hr. Mice were injected with 9 mg tamoxifen per 40 g body weight.

## Noise exposure

Mice were placed separately in wire cages inside a custom-built small wooden reverberant box, built with instructions from the Charles Lieberman lab (*Liberman and Gao, 1995*), equipped with a high-intensity speaker (Model #2446H JBL Incorporated) and amplifier (Crown). To induce maximal stereocilia core damage for experiments in which gaps were quantified or newly synthesized actin incorporation localization was analyzed, mice were exposed to 2 hr of broadband (1–22 kHz) noise at 120 dB. Following noise exposure mice were allowed to recover for 1 hr, 8 hr, or 1 week. To induce a mild permanent hearing threshold shift (*Paquette et al., 2016*), mice were exposed to 105 dB octave band (8–16 kHz) noise for 1 hr. ABRs were assessed 1 day before,1 day after, and 2 weeks after noise exposure.

## Gap quantification

Inner ears were collected and fixed prior to fine dissection of organs of Corti and utricles. For each experiment in which gaps were quantified in auditory hair cells, three images were collected for each whole-mounted cochlea at specified positions in the cochlear apex (5%, 20%, and 40% of full cochlear length). Each image contained ~15 IHCs. The percent of cells with gaps was calculated for each cochlea. For experiments in which gaps were quantified in vestibular hair cells, four images were collected for each whole-mounted utricle (two in the striola, one in the medial extrastriola, and one in the lateral extrastriola). Each extrastriolar image contained ~80 hair cells and each striolar image contained ~60 hair cells. Confocal stacks were taken at each site spanning the length of the hair bundle with a 0.40 µm step size. The number of gaps in tallest row IHC stereocilia or utricle hair bundles and the cell number was quantified for each image and the percent of cells with gaps was quantified for each utricle or cochlea.

## Cell death and stereocilia number quantification

Using the same images collected for quantification of gaps following noise exposure (three images per cochlea at defined apical regions), the number of dead IHCs was quantified to calculate the percentage of dead IHCs per cochlea. The same images were also used to quantify changes in stereocilia number following noise exposure. The number of tallest row stereocilia for each IHC was quantified, ensuring that the counted stereocilia were attached to the cuticular plate at the rootlet. The average number of stereocilia per IHC was calculated for each cochlea.

## Quantification of gap enrichment immunostaining

For vestibular hair cell quantification, both utricles were collected from each of four mice and four images were taken from each utricle (two striolar, one medial extrastriolar, and one lateral extrastriolar). For IHC quantification after noise exposure, both organs of Corti were collected 1 hr after 4 hr broadband noise exposure (described above). Three images at defined locations in the cochlear apex were taken from each organ of Corti. For each phalloidin negative gap, a line scan of fluorescence intensity was generated for the phalloidin and XIRP2, *β-actin*, γ-actin, or espin. At the center of the gap (defined by the point with lowest phalloidin fluorescence intensity) and the edge of the gap (defined as the point where phalloidin signal begins to decrease), the fluorescence intensity was determined for XIRP2, *β-actin*, γ-actin, or espin. The intensity at the center of the gap was divided by the intensity at the edge of the gap to get the enrichment ratio. The average XIRP2, *β-actin*, γ-actin, or espin gap enrichment ratio was calculated for each utricle.

## Live cell imaging

### Cell culture

Human foreskin fibroblasts (HFFs) were obtained from ATCC (CRL-252, authenticated). To minimize passage numbers, cells were banked after receiving the cells from ATCC, and experiments were performed using subcultures that were passaged less than 10 times and were less than a year old since initial purchase from ATCC. Cells are tested every 6–8 weeks for mycoplasma. HFFs were cultured in DMEM (MT10013CV, Corning) supplemented with 10% fetal bovine serum (MT35-010-CV, Corning) and 1% antibiotic-antimycotic solution (MT30004CI, Corning) at 37°C and 5% $CO_2$. At 24 hr before each experiment, HFFs were transfected with 5 μg total DNA (mApple-actin construct with EGFP-coupled versions of either full length, the LIM domain, or CTD of XIRP2) using a Neon electroporation system (Thermo Fisher Scientific) and plated on glass coverslips.

### Imaging

Cells were imaged in culture media supplemented with 20 mM HEPES (SH3023701, HyClone) at 37°C, on a Marianas Imaging System (Intelligent Imaging Innovations) consisting of an Axio Observer 7 inverted microscope (Zeiss) attached to a W1 Confocal Spinning Disk (Yokogawa) with Mesa field flattening (Intelligent Imaging Innovations), a Phasor photomanipulation unit (Intelligent Imaging Innovations), a motorized X,Y stage (ASI), and a Prime 95B sCMOS (Photometrics) camera. Illumination was provided by a TTL triggered multifiber laser launch (Intelligent Imaging Innovations) consisting of 405, 488, 561, and 637 lasers, using a 63×1.4 NA Plan-Apochromat objective (Zeiss). Temperature and humidity were maintained using a Bold Line full enclosure incubator (Oko Labs). The microscope was controlled using Slidebook 6 Software (Intelligent Imaging Innovations).

## Laser photoablation and quantitative analysis of stress fiber strain site recruitment

Induction of local stress fiber strain sites was performed as described previously (*Sala and Oakes, 2021*). Briefly, prior to initiation of a time lapse, a 5 μm linear region was drawn in Slidebook over the SFs that were to be damaged. Cells were imaged for 3 min and 30 s, alternating between the actin and XIRP2 channels, with images taken approximately every 2 s. After 30 s of imaging the steady state, the marked 5 μm region was illuminated with the 405 laser at a power of 370 μW for 1.5 s to induce a strain site via photoablation. The remainder of the time lapse was then imaged. Images were analyzed in Python. Each time lapse was first broken into two stacks representing the EGFP (XIRP2 constructs) channel and the mApple-actin channel. Each stack was first flat-field corrected and photobleach corrected (*Jost and Waters, 2019*). The EGFP-XIRP2 channel was then registered using the whole image via an efficient sub-pixel registration algorithm (*Guizar-Sicairos et al., 2008*). The calculated registration shifts from the EGFP-XIRP2 channel were then applied to the mApple-actin channel, and both channels were cropped to a region of 121×121 pixels (~21×21 μm$^2$) centered on the ablation region. For each channel, an average intensity image was created by averaging the frames in the stack before the ablation event. A relative difference was determined at each time point by subtracting the reference image from a given frame, and then dividing by the reference image. A mask of the brightest 5% of points relative to the reference image in the EGFP-XIRP2 channel was

created for each time point. The average value of the masked points in each channel was then plotted as a function of time to create a trace of the normalized fluorescence intensity for each video. Average traces were created by averaging the traces from multiple videos and plotting the mean ± the standard deviation for each time point.

## Co-immunoprecipitation

NIH-3T3 cells were obtained from ATCC (CRL-1658). To minimize passage numbers, cells were banked after receiving the cells from ATCC, and experiments were performed using subcultures that were passaged less than 10 times and less than a year old since initial purchase from ATCC. Cells are tested every 6–8 weeks for absence of mycoplasma contamination. NIH-3T3 cells were seeded on six-well plates and allowed to grow to 70% confluency at 37°C and 5% $CO_2$ in DMEM (Gibco) supplemented with 10% FBS (Life Technologies), 1 mm sodium pyruvate (Life Technologies), 100 U/ml penicillin, and 100 µg/ml streptomycin (Life Technologies). Media was exchanged with Opti-MEM (Gibco) and cells were transfected with the plasmid constructs (listed below in table) using Lipofectamine 3000 (Invitrogen) according to the manufacturer's protocols. Following 48 hr incubation, cells were lysed with NP-40 lysis buffer (50 mM Tris-HCl pH 8, 150 mM NaCl, 1% NP-40, 5% glycerol, 2 mM $MgCl_2$, 2 mM NaF, 1 mM $Na_3VO_4$, 1 mM PMSF [Thermo Scientific], and Pierce protease inhibitor cocktail [Thermo Scientific]) on ice for 10 min. Cell lysates were sonicated (550 Sonic Dismembrator, Fisher Scientific) for 2 min (with one 10 s pulse at amplitude 5 per 15 s) and centrifuged at 13,000×$g$ for 5 min. Cell lysates were added to GFP-Trap agarose beads (ChromoTek) pre-equilibrated with lysis buffer and incubated for 2 hr at 4°C. Beads were washed three times with lysis buffer with 30 mM $MgCl_2$ to depolymerize actin (*Wang et al., 2002*). Thirty µl of 2× NuPAGE LDS sample buffer (Novex by Life Technologies) was added to the samples. Ten µl of each sample and 3% input of each cell lysate was loaded on a 4–12% Bis-Tris pre-cast SDS-PAGE gel (Invitrogen) and transferred to a PVDF membrane (Bio-Rad). The membrane was blocked with Odyssey Blocking Buffer (LI-COR) for 30 min at RT and incubated with the following primary antibodies at 4°C overnight in TBST with 5% BSA: monoclonal mouse GFP antibody (B-2: sc-9996, Santa Cruz Biotechnology, Inc, 1:500), polyclonal pan-XIRP2 antibody (custom generated [see above], Proteintech, 1:1000), monoclonal mouse $IgG_1γ$-actin antibody (1–24: sc-65635, Santa Cruz Biotechnology, Inc, 1:1000), and mouse monoclonal β-actin antibody (AC-15: sc-69879, Santa Cruz Biotechnology, Inc, 1:1000). Membranes were washed three times for 5 min with TBST and then incubated with the corresponding secondary antibodies (donkey anti-mouse IgG or donkey anti-rabbit IgG-conjugated HRP [Jackson ImmunoResearch]) for 1 hr RT at 1:10,000 in TBST with 5% non-fat milk. Membranes were then washed three times for 4 min with TBST and treated with Pierce ECL Western Blotting Substrate (Thermo Scientific) according to the manufacturer's protocol. Membranes were imaged using the ImageQuant LAS system (GE Science). Sequence information of XIRP2 mutant constructs is as follows.

## DNA constructs

### XIRP2 mutant constructs (in pEGFP-C1 backbone)

| XIRP2 constructs | | Sequence |
|---|---|---|
| NTD (incl. LIM) | Forward primer | GGATCCACCGGATCTAGATAACTGATCATAATCAGCCATACC |
| | Reverse primer | AGACTTAAAAAGTTGTTTAAAATGAGGCTTGCAGTATATTCGTCC |
| | Resulting peptide sequence | MADNLEPSTFPIQKGSLSRLRQKWESSDCQRNESYPGGSRCKLFQL KESNLLEPEGEALLTLDPPEAPSLPCCKKEEVLCGEPEDGVPEDQND KLKDCGQPEVLKEDSLTGRRRIERFSIALDELRSIFESPKSSINSAGPV EYVQKEVEIGRSLCSPTFRSLPGSHADDFMRDSDLKSEEVPLDKMSP KSGQSPSMEVTFNLTKPGDVSTAGSEDQSDLLEALSLKERMARYQAA VSRGDTRSFSANVMEESDVCTVPGGLAKMKRQFEKDKMTSTCNAFSE YQYRHESRAEQEAIHSSQEIIRRNEQEVSKGHGTDVFKAEMMSHLEK HTEETKQASQFHQYVQETVIDSPEEEELPKVSTKILKEQFEKSAQENF LRSDKETSPPAKCMKKLLVQDKEICIICQKTVYPMECLIADKQNFHKS CFRCHHCSSKLSLGNYASLHGRIYCKPHFKQLFKS |
| CTD | Forward primer | TTTGGACACAAGCAACATAAAGATCGATGGAATTGCAAAAAC |
| | Reverse primer | AGCTCGAGATCTGAGTCCGGACTTGTACAGCTCGT |
| | Resulting peptide sequence | KGNYDEGFGHKQHKDRWNCKNQSSLVDFIPSGEPDAHENPT ADTLLLGDLTTHPDACNSKRQDNDLRKWGDRGKLKIVWPPC QEMPKKNSPPEEEFKVNKAKWPPEVTIPVPSEFKRESLTEHVK TLESQGQEQDSVPDLQPCKHVCQKEDITGIKEIKGYEERKDEK EAKDTLKDAEGLRSKRKSGMEFNDHNAHAQSDGKEKNALVN EADSADVLQVANTDDEGGPENHRENFNNNNNNNSVAVSSLN NGRRKISISERPRLLQAVSEANYYTSEYQIKNFNNASKISELLGI FESQKLSSKKVLALALERTADRGTAGSPLQLVLEPGLQQGFSVK GENLAASPDVSPLHIKGNHENNKNVHLFFSNTVKITSFSKKHNI LGCDVMDSVDQLKNMSCLYLRELGKNVKCWHGETAGAARHG GKMCFDAQSQGSAAKPVFPSMQCQTQHLTVEEQIKRDRCYSDSEAD* |
| LIM only (generated from NTD) | Forward primer | GAAATTTGCATAATTTGTCAAAAGACAGTTTATCCAATGGAGTGC |
| | Reverse primer | AGCTCGAGATCTGAGTCCGGACTTGTACAGCTCGT |
| | Resulting peptide sequence | EICIICQKTVYPMECLIADKQNFHKSCFRCHHCSSKLSLGNYASLHGRIYCKPHFKQLFKS |

### HA-γ-actin WT and HA-γ-actin G13R constructs (in pcDNA3.1 backbone)

The HA-γ-actin WT-pcDNA3.1 was generated by VectorBuilder Inc (Chicago, IL, USA). The G13R mutant was generated by site-directed mutagenesis (Q5 Site-directed mutagenesis kit, NEB) of the WT construct. Primer sequence for mutagenesis: γ-actin G13R forward: TCGTCATTGACAATCGCTCC GGCATGTGCA. γ-Actin G13R reverse: TGCACATGCCGGAGCGATTGTCAATGACGA. Resulting amino acid sequence (first 20 aa): γ-actin WT: MEEEIAALVIDNGSGMCKAG. γ-Actin G13R: MEEE-IAALVIDNRSGMCKAG. In red indicated is the mutated amino acid.

## Statistical analysis

For statistical analysis, GraphPad Prism (La Jolla, CA, USA) and Microsoft Excel were used. Two-way ANOVA was used to determine statistically significant differences in the ABR and threshold changes following noise exposure recovery analyses. Post hoc analysis (Tukey's multiple comparisons test) was used for comparisons between genotypes at individual frequencies. Two-tailed, unpaired Student's $t$ tests were used for comparisons between two groups, and one-way ANOVA was used for comparisons between three or more groups, with Tukey's multiple comparisons tests to determine statistical significance between two individual groups. p-Values smaller than 0.05 were considered statistically

significant, other values were considered not significant (n.s.). Asterisks in the figures indicate p-values (*p < 0.05, **p < 0.01, and ***p < 0.001). All error bars indicate standard error of the mean (SEM). Sample sizes were determined based on variance from previous experiments in the lab for ABRs. For other experiments, sample size was determined from variance observed from pilot experiments. Whenever possible, quantifications were performed in a blinded manner.

## Acknowledgements

We would like to thank Alban Gaultier for providing the Ubc-Cre^ERT2 mouse line. This study was supported by NIH/NIDCD grants R01DC014254, R56DC017724, and R01DC018842 (to J.-B.S.), R01DC011835 (to K.B.A.), and 1F31DC017370-01 (to E.L.W.). Further support to J.-B.S. was provided by the Virginia Lions Hearing Foundation (VLHF) and the Owens Family Foundation. P.W.O. was supported by an National Science Foundation CAREER Award (#2000554).

## Additional information

### Funding

| Funder | Grant reference number | Author |
|---|---|---|
| National Institute on Deafness and Other Communication Disorders | R01DC014254 | Elizabeth L Wagner<br>Jun-Sub Im<br>Sihan Li<br>Jung-Bum Shin |
| National Institute on Deafness and Other Communication Disorders | R01DC018842 | Elizabeth L Wagner<br>Jun-Sub Im<br>Jung-Bum Shin<br>Sihan Li |
| National Institute on Deafness and Other Communication Disorders | F31DC017370 | Elizabeth L Wagner |
| National Institute on Deafness and Other Communication Disorders | R01DC011835 | Karen B Avraham |
| National Science Foundation | 2000554 | Patrick W Oakes |
| Owens Family Foundation | | Jung-Bum Shin |
| National Institute on Deafness and Other Communication Disorders | R56DC017724 | Elizabeth L Wagner<br>Jun-Sub Im<br>Jung-Bum Shin<br>Sihan Li |
| National Institute on Deafness and Other Communication Disorders | 1F31DC017370-01 | Elizabeth L Wagner |
| Virginia Lions Hearing Foundation | | Jung-Bum Shin |
| Owens Family Foundation | | Jung-Bum Shin |

The funders had no role in study design, data collection and interpretation, or the decision to submit the work for publication.

### Author contributions

Elizabeth L Wagner, Conceptualization, Data curation, Formal analysis, Funding acquisition, Investigation, Visualization, Methodology, Writing – original draft, Writing – review and editing; Jun-Sub Im, Investigation, Visualization, Methodology; Stefano Sala, Formal analysis, Investigation; Maura I Nakahata, Terence E Imbery, Daniel Chen, Investigation; Sihan Li, Investigation, Visualization, Writing – review and editing; Katherine Nimchuk, Investigation, Writing – review and editing; Yael Noy, Wenhao

Xu, Resources, Methodology; David W Archer, George Hashisaki, Resources; Karen B Avraham, Formal analysis, Funding acquisition, Writing – original draft, Writing – review and editing; Patrick W Oakes, Conceptualization, Data curation, Formal analysis, Supervision, Funding acquisition, Investigation, Writing – review and editing; Jung-Bum Shin, Conceptualization, Resources, Data curation, Formal analysis, Supervision, Funding acquisition, Validation, Investigation, Methodology, Writing – original draft, Project administration, Writing – review and editing

## Author ORCIDs
Yael Noy http://orcid.org/0000-0002-7425-5669
Patrick W Oakes http://orcid.org/0000-0001-9951-1318
Jung-Bum Shin http://orcid.org/0000-0003-3047-0874

## Ethics
Informed consent and consent to publish was not necessary because the human utricles used in this study were obtained as a byproduct of a surgical labyrinthectomy. Patient information was de-identified.

The care and use of animals for all experiments described conformed to NIH guidelines. Experimental mice were housed with a 12:12h light:dark cycle with free access to chow and water, in standard laboratory cages located in a temperature and humidity-controlled vivarium. The protocols for care and use of animals (3822) was approved by the Institutional Animal Care and Use Committees at the University of Virginia, which is accredited by the American Association for the Accreditation of Laboratory Animal Care. All procedures involving Elmod1 knockout animals met the guidelines described in the National Institutes of Health Guide for the Care and Use of Laboratory Animals and was approved by the Animal Care and Use Committees of Tel Aviv University (M-12-046).

## Decision letter and Author response
Decision letter https://doi.org/10.7554/eLife.72681.sa1
Author response https://doi.org/10.7554/eLife.72681.sa2

# Additional files

## Supplementary files
• Transparent reporting form

## Data availability
All data generated or analysed during this study are included in the manuscript and supporting file.

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
