## [Editor Report]

Hair cells, the sensory receptors of the inner ear, are easily damaged by exposure to noise. This study provides mechanistic insight into the process by which the mechanosensitive stereocilia of the hair cells can recover from damage-induced gaps in their actin core, possibly allowing for the rescue of transient hearing loss after noise exposure. This manuscript will be of considerable interest to the inner ear field as well as readers with a broader interest in actin cytoskeleton dynamics.

---

## [Decision Letter]

**Decision letter after peer review:**

Thank you for submitting your article "Repair of Noise-Induced Damage to Stereocilia F-actin Cores is Facilitated by XIRP2" for consideration by *eLife*. Your article has been reviewed by 3 peer reviewers, and the evaluation has been overseen by a Reviewing Editor and Andrew King as the Senior Editor. The following individuals involved in review of your submission have agreed to reveal their identity: Gregory M Alushin (Reviewer #1).

Essential revisions:

1) All the reviewers stated that additional experiments are needed to identify the mechanism of action of XIRP2. In particular, the claim made that XIRP2 likely employs a mechanism similar to other LIM domain-containing proteins for repairing mechanical damage to actin bundles is not well supported by the data. Evidence should be provided to show that XIRP2 LIM domains bind to stressed, strained or damaged actin and that XIRP2 recruits G-actin to breaks in noise-damaged stereocilia (and that this process is absent in XIRP2 knock-out mice). Please see the recommendations to the authors in each review for specific suggestions for how this key point might be addressed.

2) A concern was raised about the specificity of the antibody used for the G-actin immunolabeling and whether this has been validated previously.

3) More detailed characterization of the stereocilia in Xirp2-∆Cterm mice is needed, as set out in more detail in the reviews.

4) A number of important experimental details were felt to be missing from the manuscript and an important query raised over the statistical tests used.

*Reviewer #1 (Recommendations for the authors):*

The paper could be strengthened by additional data directly probing the function of short XIRP2's LIM domain without deleting the entire C-terminal half of the protein, as well as monitoring localization of ectopically expressed XIRP2 in tissue culture cells to stress fiber repair sites which have previously been found to recruit other mechano-responsive LIM proteins, including the repair factor zyxin. We nevertheless believe this story overall represents a satisfactory advance suitable for publication in *eLife*, pending revision to address the major and minor concerns outlined below:

1) The authors make a significant analogy throughout the paper of XIRP2-mediated repair to zyxin/paxillin-mediated stress fiber repair, proposing that it's LIM domain-containing isoform may recognized tensed actin. Almost all mechano-responsive families of LIM domain proteins contain multiple tandem LIM domains, which bind to tensed actin at Stress Fiber Strain Sites (SFSS) by a cumulative mechanism (Sun et al., PMID 33058779; Winkelman et al., PMID 32989126). This activity is clearly conserved, as a yeast LIM domain protein homologous to paxillin localized to SFSS in mammalian cells (Winkelman et al.)

The authors' sequence analysis only identified a single predicted LIM domain in the short isoform of XIRP2, which would be incompatible with the requirement for tandem LIM domains. However, Sala and Oakes (PMID 34038160) recently reported that individual LIM domains excised from the protein testin rapidly localize to SFSS, which was also observed for the protein PDLIM1 by Sun et al., albeit with substantially delayed kinetics, suggesting recognition of a mark other than strained actin.

Based on the current evidence in the literature, it is thus unclear whether XIRP1 is likely to be using a similar mechanism to strained actin binding by tandem LIM proteins. The authors could address this point by expressing fluorescently labeled short-isoform XIRP2 in tissue culture cells (e.g. Mouse Embroynic Fibroblasts or U2OS cells), and observing whether it localizes to SFSS marked by co-expression with zyxin. This would substantially strengthen the authors case for a strained-actin binding mechanism.

2) By introducing a STOP codon before the LIM domain of short XIRP2 in mice IHCs, the authors demonstrate that this truncated variant of short XIRP2 acts as a separation of function mutant that phenocopies Xirp2 knockout mice in γ-actin recruitment deficiency, but still localizes to stereocilia F-actin cores. The authors then proceed to claim that this is consistent with previously reported LIM domain proteins recognizing tensed actin. Based on the sequence diagram of short XIRP2 in Figure 6. A-C, the authors' XIRP2-ΔCterm construct appears to truncate over half of short XIRP2. While the authors address the alternative hypothesis that the non-LIM domain-containing C-terminus could also recognize gaps in stereocilia, I am concerned about the choice to truncate over half of the protein as a method to ablate the LIM domain of short XIRP2 in vivo. The authors could demonstrate that the LIM domain of short XIRP2 is required for gap localization/repair by overexpressing either WT short XIRP2, or ΔLIM short XIRP2 in XIRP2-ΔCterm mice IHCs. While this is not endogenous expression, it overcomes the challenge of introducing mutations also impact the long isoform. If this is not technically feasible, ectopic expression in tissue culture cells and monitoring SFSS localization as described above would be a suitable alternative.

*Reviewer #2 (Recommendations for the authors):*

1) Gamma-actin immunolabeling is unexpectedly absent from stereocilia shaft (Figure 5A, C; Figure 6J). However, to my knowledge, previous labeling of both β- and γ-actin showed signal along the whole stereocilium (Belyantseva PNAS 2009, or for example Zhang Nature 2012). The antibody used appears to be a Santa Cruz commercial reagent. Was it validated previously, and is there an explanation for why it would not label F-actin? As this antibody appears different from the one(s) used previously, could it detect only monomeric actin? If this is the case, how was this conclusion established? These questions are pertinent as this product is used to generate image panels and quantifications that are directly relevant to the main conclusions of the study.

2) The Xirp2 DelCter strain is a very elegant way to address the specific role of the short isoform and the LIM domain during gap repair. This strain has the powerful potential to directly link gap repair to ABR threshold shits, as the truncated DelCter short isoform protein is still at stereocilia shafts, but missing at gaps, and the long isoform is intact. However key results are missing to verify that DelCter animals indeed have an excess of gaps, as the full KO, and as expected if the short isoform is the protein enriched there via the LIM domain. Currently, the DelCter strain is only used to show loss of XIRP2 and γ-actin enrichment at gaps (Figure 6). Along the same lines, the TTS rescue in Figure 7 would be more powerful if it included the DelCter and not (only) the full KO strain. This would ideally substantiate the claim that poorer threshold recovery in Xirp2 mutant(s) is linked to lack of repair at gaps.

3) Experimental details are generally limited in the current manuscript on several levels. This is reflected both in figure legends and Methods sections.

The number of cells analyzed is often not clear, and in some cases, other unexpected units are used instead. For example, in Figure 3J, n (=21) is "images" and there appears to be 21 points on the graph, but how exactly is "enriched sites per HC" (Y axis) derived from one image? This also applies to Figure 1D-G, Figure 4 and others. In some cases, n is a number of organ of Corti, but the uncertainty is the same, as Y axis represents a % of cells, of # gaps per cell. It would be best to use n as the total number of cells analyzed while keeping information about organ and animal number. In addition, the corresponding analysis method(s) to derive Y values from images could be more detailed in Methods.

In the text, gap quantification changes from #gaps/hair cell (line 87: 0.08 before noise) to % of HC with gaps (line 94: 4.89% before noise). If there is a good rationale to switch between these two formulations, it would be useful to clarify and discuss why. Do rare HCs with gaps have more than one gap? Does it mean that having one gap makes a HC more susceptible to get more?

According to Methods, statistical tests are either t-tests for paired comparisons or 2-way ANOVA for ABRs. I do not pretend to be an expert, but many graphs in the study compare 3 or 4 conditions, for example "no noise" with 1h post noise and 1-week post-noise (e.g Figure 2J, Figure 4E-G and others). I wonder if one-way ANOVA with multiple comparisons would be more appropriate and stringent in these cases. For ABR graphs, it would be preferable to have multiple p values that indicate significance by frequency, as in some cases significance must differ (Figure 6L; Figure 7). This can be done with a multiple comparison post-hoc test like Sidak.

There is no supplementary material in the current manuscript, and it is likely that data generated already might be useful to bolster the conclusion of the study and make it more transparent. That includes negative results when obtained, or validation and additional characterization of the mouse strains generated in this study, especially the powerful Xirp2 DelCter, for example.

Methods include a de-noising process, but its goal is not stated. Was it used to visualize gaps? In general, a section about image acquisition and processing might be useful. How were samples mounted and imaged to quantify gaps? Were confocal stacks taken? Are images projections from stacks?

4) The hearing results would benefit from a more detailed discussion. Compared to C57BL/6J, the CBA background does not show threshold shifts at 11-12 week, and shows much milder shifts at 20 and 40 week (Figure 7). This suggests that early hearing defects in the full KO are largely caused by a modifier locus in C57BL/6J, and also suggests that threshold shifts in the DelCter strain at 20 month ( Figure 6L) are unlikely to be caused by defective short XIRP2 protein alone.

Other comments:

Figure 2A, B: the phalloidin and XIRPs panels must have been switched by mistake for the utricle

Figure 1I: the tamoxifen caption looks wrong "2x <;:;Tamoxifen"?

Figure H: please use same terminology for Β-actin-Egfp throughout the text and figures. I think the Egfp is C-terminal, so GFP-Actin in H is imprecise, and does not match panels K-L.

Figure 4A, C: captions to indicate that these are Xirp2 mutants would be helpful. In E-F, indicating that plots refer to the cochlea would be useful too.

Figure 5E: what does γ-actin gap ratio mean? An explanation is missing in the legend.

Figure 6G: please confirm that phalloidin and XIRP2 panels are not switched here as well, because phalloidin shows slight over-enrichment at tips, as XIRP2 in previous panels.

Line 99: "To address the possibility that the decreased percentage of IHCs with gaps was not due to the death of..". Change for "was due to the…".

Line 111: please cite the Narayanan reference when the Actb-Egfp strain is first introduced.

Line 137: I do not think there is a mention of stereocilia gaps in Elmod1 mutants in Krey 2018 although other defects are detailed? This could be clarified. How did this strain become an "actin gap" model? Ptprq1 actin gaps had been reported previously by contrast.

Line 138: the context of human samples would be valuable to add, if possible. Are these aged, or noise-damaged samples?

Line 221: hearing loss from ABR threshold shifts in DelC strain. Please indicate stage and strain (C57BL/6J) in the legend at least, or better, in a caption as in Figure 7.

Line 246: A 105dB for 30min regime and the context makes me wonder whether the text should read ".… background to TTS-inducing noise", and not " to PTS-inducing noise"? Did wild-type controls fully recover in this experiment?

Line 268: remove one occurrence of "contribution".

Line 293: “Combined with the decrease in gaps we found following noise exposure…” Technically, there is an increase in gaps following noise exposure, so a more precise formulation would be “..we found after recovery from noise exposure”.

Line 298: besides possibly explaining the low frequency of Actb-Egfp induction, recycling of actin monomers may principally explain the surprising low intensity of Egfp signal at “repaired” gaps, at least compared to stereocilia tips.

Line 607: remove double period.

*Reviewer #3 (Recommendations for the authors):*

One key feature of the proposed mechanism is that monomeric γ actin is recruited to breaks by XIRP2. I have three concerns.

– The first is that there is very little evidence that XIRP2 can bind to G-actin. Although actin immunoprecipitates with GFP-XIRP2, the association could be indirect. In addition, washing with 30 mM MgCl2 is not a common way of disassembling F-actin. Are there additional references using this approach or experimental evidence of effectiveness? Even so, it would still be possible that F-actin bound to XIRP2 is resistant to disassembly. A better approach would be to show direct binding of purified XIRP2 to G-actin using an in vitro binding assay.

– One strength of the noise damage approach is that breaks are clearly shown to be induced, so they don't just result from damage during dissection. The same cannot be said for breaks in utricular stereocilia, which are long and might easily be broken during dissection. Is it possible that XIRP2 mutant stereocilia are more fragile? In that case breaks lacking γ-actin may have arisen when tissue was dissected and fixed before the normal repair complex was recruited. The idea that XIRP2 recruits G-actin to breaks would be more compelling if this were also deficient in XIRP2 KO IHC stereocilia, particularly a week after noise damage when the number of breaks is increased from baseline.

– The loss of γ-actin at breaks in mutant XIRP2 stereocilia could be indirect, stemming from changes to the structure of the stereocilia core (known to be disordered in the KO) and not the loss of XIRP2 at breaks. The δ-C-term mutant mouse is a nice approach that may well answer this question, but this mouse needs additional characterization before it is possible to interpret whether loss of γ-actin or mutant XIRP2 from breaks is due to loss of the LIM domains or just defects in the core. Characterization could include comparing stereocilia height, width, and possibly F-actin organization to WT and KO mice. The more critical experiment is measuring the number of breaks in XIRP2 δ-C-term IHC stereocilia before and after noise, and assessing γ-actin levels in those IHC stereocilia breaks.

The second part of the hypothesis is that XIRP2 is recruited to breaks via its LIM domains. The description of SFSS and LIM domains in the introduction was such an interesting comparison that it was disappointing not to see a direct test of whether XIRP2 LIM domains bind to stressed, strained, or damaged F-actin. Along those lines, it would be interesting if LIM domains from other proteins that do bind strained actin would detect stereocilia breaks when used as probes. Either finding would strengthen the paper considerably, though both are challenging experiments.

[Editors' note: further revisions were suggested prior to acceptance, as described below.]

Thank you for resubmitting your work entitled "Repair of Noise-Induced Damage to Stereocilia F-actin Cores is Facilitated by XIRP2 and its Novel Mechanosensor Domain" for further consideration by *eLife*. Your revised article has been evaluated by Andrew King (Senior Editor) and a Reviewing Editor.

The manuscript has been improved but there are some remaining minor issues that need to be addressed, as outlined below in the comments from Reviewer #1, before we can make a final decision:

*Reviewer #1 (Recommendations for the authors):*

In the revised manuscript, the authors conducted additional experiments to support their claims that XIRP2 (1) is mechanoresponsive like other LIM domain-containing proteins but uses a distinct non-LIM mediated localization mechanism, and (2) binds monomeric γ-actin. Strikingly, their laser ablation experiments in fibroblasts revealed that it is not the LIM domain of XIRP2 that is mechanoresponsive to laser-induced stress fiber strain sites, but rather the structurally uncharacterized C-terminal domain (CTD). Their new results support a mechanism by which XIRP2 is recruited to sites of damage within F-actin core through its CTD to mediate repair in inner hair cells (IHCs). The additional data substantially strengthen their paper and reveal new insights into the mechanism of XIRP2-mediated repair of stereocilia. Furthermore, the authors' identification of a mechanism for rapid recruitment to mechanically damaged actin structures that does not depend on LIM domains is a very exciting development that is sure to stimulate additional work in the field.

We believe the authors' revisions have addressed our previous concerns and the paper is now suitable for publication in *eLife*, pending minor textual revision to address the points below:

1. In line 431, the authors include vinculin as one of the proteins modified ("tuned") by mechanical forces on F-actin, citing Mei et al. (2020). In that paper, only ɑ-catenin was observed to display force-activated F-actin binding, while vinculin was found to be insensitive to forces along F-actin. Thus, the authors should not include vinculin in this sentence.

2. Lines 426-451: In a generally balanced fashion, the authors write that future work is necessary to determine the exact mechanosensitive domain in XIRP2's C terminus and the structural features that mediate its force-dependent interactions with F-actin. Aligned with this, we note that to our knowledge neither the individual LIM domains of testin nor XIRP2's force-responsive domain have been shown to directly bind tensed F-actin, which would require in vitro experiments with purified proteins. Some of the language in the discussion implies that this is the case. While it is of course very reasonable to speculate that this is true, we suggest that the authors be careful with their wording to make clear that this is currently a (leading) hypothesis.

2.

---

## [Author Response]

Essential revisions:1) All the reviewers stated that additional experiments are needed to identify the mechanism of action of XIRP2. In particular, the claim made that XIRP2 likely employs a mechanism similar to other LIM domain-containing proteins for repairing mechanical damage to actin bundles is not well supported by the data. Evidence should be provided to show that XIRP2 LIM domains bind to stressed, strained or damaged actin and that XIRP2 recruits G-actin to breaks in noise-damaged stereocilia (and that this process is absent in XIRP2 knock-out mice). Please see the recommendations to the authors in each review for specific suggestions for how this key point might be addressed.

We considered this the key experiment, and we are happy to report that we now provide direct evidence, through a collaboration with an expert in mechanobiology (Patrick Oakes), that XIRP2 is recruited to strained actin sites (details are provided below).

2) A concern was raised about the specificity of the antibody used for the G-actin immunolabeling and whether this has been validated previously.

These concerns were addressed (details are provided below).

3) More detailed characterization of the stereocilia in Xirp2-∆Cterm mice is needed, as set out in more detail in the reviews.

Partly addressed (details are provided below).

4) A number of important experimental details were felt to be missing from the manuscript and an important query raised over the statistical tests used.

Addressed (details are provided below).

Reviewer #1 (Recommendations for the authors):The paper could be strengthened by additional data directly probing the function of short XIRP2's LIM domain without deleting the entire C-terminal half of the protein, as well as monitoring localization of ectopically expressed XIRP2 in tissue culture cells to stress fiber repair sites which have previously been found to recruit other mechano-responsive LIM proteins, including the repair factor zyxin. We nevertheless believe this story overall represents a satisfactory advance suitable for publication in eLife, pending revision to address the major and minor concerns outlined below:1) The authors make a significant analogy throughout the paper of XIRP2-mediated repair to zyxin/paxillin-mediated stress fiber repair, proposing that it's LIM domain-containing isoform may recognized tensed actin. Almost all mechano-responsive families of LIM domain proteins contain multiple tandem LIM domains, which bind to tensed actin at Stress Fiber Strain Sites (SFSS) by a cumulative mechanism (Sun et al., PMID 33058779; Winkelman et al., PMID 32989126). This activity is clearly conserved, as a yeast LIM domain protein homologous to paxillin localized to SFSS in mammalian cells (Winkelman et al.)The authors' sequence analysis only identified a single predicted LIM domain in the short isoform of XIRP2, which would be incompatible with the requirement for tandem LIM domains. However, Sala and Oakes (PMID 34038160) recently reported that individual LIM domains excised from the protein testin rapidly localize to SFSS, which was also observed for the protein PDLIM1 by Sun et al., albeit with substantially delayed kinetics, suggesting recognition of a mark other than strained actin.Based on the current evidence in the literature, it is thus unclear whether XIRP1 is likely to be using a similar mechanism to strained actin binding by tandem LIM proteins. The authors could address this point by expressing fluorescently labeled short-isoform XIRP2 in tissue culture cells (e.g. Mouse Embroynic Fibroblasts or U2OS cells), and observing whether it localizes to SFSS marked by co-expression with zyxin. This would substantially strengthen the authors case for a strained-actin binding mechanism.

We thank Donovan Phua and Dr. Alushin, who identified themselves as the reviewers, for suggesting the laser ablation experiment. We collaborated with Patrick Oakes’ lab to conduct the experiments, and the results fully support our hypothesis that XIRP2 is recruited to strained actin sites in a force-dependent manner. Surprisingly, the recruitment to laser-induced stress fiber strain sites was independent from the predicted LIM domain, but dependent on the uncharacterized C-terminal domain (CTD) of XIRP2. This marks the discovery of a novel mechanosensor domain. As Dr. Alushin writes in his recent review, the interaction of a variety of proteins with F-actin is tuned by strain, but so far, only a subset of LIM domains were ascribed a on-off switch-like recruitment to strained actin. Our data show that the CTD of XIRP2 behaves in such manner. Interestingly, the mechanosensitivity of the C-terminal domain is masked in the full length XIRP2, suggestive of a yet unknown regulatory mechanism. In addition to its relevance for hair cell biology, the discovery of this domain will help better understand the molecular and structural basis of F-actin’s mechanosensitivity.

The additional data support our proposed model of stereocilia F-actin repair, according to which mechanical stress (induced e.g. by noise) creates a partial depolymerization of F-actin that leaves the remaining actin fibers under increased strain. XIRP2, through its CTD, is then recruited to the lesion in a force-dependent manner and mediates its repair.

2) By introducing a STOP codon before the LIM domain of short XIRP2 in mice IHCs, the authors demonstrate that this truncated variant of short XIRP2 acts as a separation of function mutant that phenocopies Xirp2 knockout mice in γ-actin recruitment deficiency, but still localizes to stereocilia F-actin cores. The authors then proceed to claim that this is consistent with previously reported LIM domain proteins recognizing tensed actin. Based on the sequence diagram of short XIRP2 in Figure 6. A-C, the authors' XIRP2-ΔCterm construct appears to truncate over half of short XIRP2. While the authors address the alternative hypothesis that the non-LIM domain-containing C-terminus could also recognize gaps in stereocilia, I am concerned about the choice to truncate over half of the protein as a method to ablate the LIM domain of short XIRP2 in vivo. The authors could demonstrate that the LIM domain of short XIRP2 is required for gap localization/repair by overexpressing either WT short XIRP2, or ΔLIM short XIRP2 in XIRP2-ΔCterm mice IHCs. While this is not endogenous expression, it overcomes the challenge of introducing mutations also impact the long isoform. If this is not technically feasible, ectopic expression in tissue culture cells and monitoring SFSS localization as described above would be a suitable alternative.

In light of our finding that the C-terminal domain, and not the LIM domain of XIRP2, is responsible for recruitment to strained actin, we believe that the phenotype found in the *XIRP2-ΔCterm* mice (now called *Xirp2-ΔLIM/CTD* mouse*)* lacking the LIM and CTD, fully supports our conclusions. Although technically possible with gene gun transfection, transfecting hair cells with overexpressed mutant constructs is not very efficient and overexpression artifacts are expected. Future experiments are planned to further narrow down the mechanosensitive peptide in the CTD, and to manipulate this minimal mechanosensor domain using a more targeted mutant mouse line.

Reviewer #2 (Recommendations for the authors):1) Gamma-actin immunolabeling is unexpectedly absent from stereocilia shaft (Figure 5A, C; Figure 6J). However, to my knowledge, previous labeling of both β- and γ-actin showed signal along the whole stereocilium (Belyantseva PNAS 2009, or for example Zhang Nature 2012). The antibody used appears to be a Santa Cruz commercial reagent. Was it validated previously, and is there an explanation for why it would not label F-actin? As this antibody appears different from the one(s) used previously, could it detect only monomeric actin? If this is the case, how was this conclusion established? These questions are pertinent as this product is used to generate image panels and quantifications that are directly relevant to the main conclusions of the study.

The antibody used in our study has been previously validated in hair cells (Andrade, 2015, “Evidence for changes in β- and γ-actin proportions during inner ear hair cell life”). This paper also showed images where the antibody predominantly labeled monomeric γ-actin at gaps. They suggested that the lack of labeling of the rest of the stereocilia core could be due to poor penetration of the primary and secondary antibodies into the highly compact F-actin.

Additionally, the observed staining pattern with this antibody appears very similar to those used in Belyantseva PNAS 2009. Like was shown in this paper, we do see labeling of whole stereocilia in some hair cells. However, in most cases, the antibody primarily labels enriched staining in gaps, with minimal staining along the rest of the stereocilium. Although not discussed in Belyantseva 2009, their antibody also appeared to predominantly label gaps in some vestibular hair cells, with weak/absent staining in the rest of the bundle, pictured in their Figure 3A-F and described here:

“In the course of characterizing the localization of the cytoplasmic actins, we observed occasional gaps in phalloidin staining of F-actin cores of vestibular hair cell stereocilia (Figure 3A–C). The gaps were most frequently observed at the base and along the length of stereocilia in the tallest row (Figure 3D). Using our γ_cyto_-actin specific antibodies, which recognize both globular (G) and filamentous (F) forms of actin (see SI Text), we found that gaps were enriched in γ_cyto_-actin. This actin population is likely to be predominantly monomeric, because phalloidin recognizes only filamentous actin (Figure 3 A–D). Usually gap staining was much more intense relative to that along the length of stereocilium (Figures 3 A–D and 3 F–M), which may be caused by enhanced antibody accessibility within the gaps. Alternatively, intense gap staining could be partially explained by the redistribution of γ_cyto_-actin to F-actin gaps from a pool of available non-filamentous actin within a stereocilium. A similar redistribution to F-actin gaps was also seen for β_cyto_-actin (Figure S3). It is likely that β_cyto_-actin is also recruited to the gaps from a pool of non-filamentous actin, as staining intensity along a stereocilium with a gap was not different from the intensity of staining along a stereocilium without gaps (Figure S3B). The same pattern of staining was also observed for DNase I (Figure 3E), a marker for monomeric actin (G-actin) (21), and espin (Figure 3F), an actin bundling protein essential for stereocilia formation, which is reported to have both F- and G-actin binding sites (22).”

Like they describe, we presume that the antibody is predominantly labeling monomeric γ-actin because of the lack of phalloidin staining at these sites. To support this conclusion, we co-labeled with Alexa Fluor 488-DNaseI, which binds specifically to monomeric actin (see Figure 5A).

2) The Xirp2 DelCter strain is a very elegant way to address the specific role of the short isoform and the LIM domain during gap repair. This strain has the powerful potential to directly link gap repair to ABR threshold shits, as the truncated DelCter short isoform protein is still at stereocilia shafts, but missing at gaps, and the long isoform is intact. However key results are missing to verify that DelCter animals indeed have an excess of gaps, as the full KO, and as expected if the short isoform is the protein enriched there via the LIM domain. Currently, the DelCter strain is only used to show loss of XIRP2 and γ-actin enrichment at gaps (Figure 6). Along the same lines, the TTS rescue in Figure 7 would be more powerful if it included the DelCter and not (only) the full KO strain. This would ideally substantiate the claim that poorer threshold recovery in Xirp2 mutant(s) is linked to lack of repair at gaps.

We agree that further characterization of the XIRP2-∆Cterm strain (now called *Xirp2-*Δ*LIM/CTD mouse)* would strengthen our study. As suggested, we quantified the number of gaps in the XIRP2-∆Cterm mice compared to WT and found an increase in gaps in both the utricle and cochlea. However, although it would greatly strengthen our evidence for gap repair being important for hearing threshold recovery, it would take substantially more time to obtain the mice needed for this experiment, because the WT and XIRP2 KO mice used in the TTS experiment in Figure 7 were backcrossed to the CBA/J background and we are currently only maintaining the XIRP2-∆Cterm mice on a C57Bl/6J background.

3) Experimental details are generally limited in the current manuscript on several levels. This is reflected both in figure legends and Methods sections.The number of cells analyzed is often not clear, and in some cases, other unexpected units are used instead. For example, in Figure 3J, n (=21) is "images" and there appears to be 21 points on the graph, but how exactly is "enriched sites per HC" (Y axis) derived from one image? This also applies to Figure 1D-G, Figure 4 and others. In some cases, n is a number of organ of Corti, but the uncertainty is the same, as Y axis represents a % of cells, of # gaps per cell. It would be best to use n as the total number of cells analyzed while keeping information about organ and animal number. In addition, the corresponding analysis method(s) to derive Y values from images could be more detailed in Methods.

All quantification of gap number has been changed to use consistent units of % of cells with gaps and all data points have been updated to reflect n= organs (utricle or organ of Corti). As described in the Methods section, for each organ of Corti, 3 images were taken from each organ at defined locations with ~15 IHCs each, for a total of ~45 cells per organ. For each utricle, 4 images were taken at defined locations with ~80 cells per striola image and ~60 cells per extrastriola image for a total of ~280 cells per organ.

In the text, gap quantification changes from #gaps/hair cell (line 87: 0.08 before noise) to % of HC with gaps (line 94: 4.89% before noise). If there is a good rationale to switch between these two formulations, it would be useful to clarify and discuss why. Do rare HCs with gaps have more than one gap? Does it mean that having one gap makes a HC more susceptible to get more?

There was not a specific rationale for switching between methods of quantification. To provide consistency, all instances of gap prevalence quantification are now reported as % of cells with gaps. It does seem to be the case that having at least one gap may make cells more susceptible to getting more. To reduce the effect of outlier cells (those with >3 gaps) on the data, we chose to quantify % of cells with gaps, rather than gaps per cell.

According to Methods, statistical tests are either t-tests for paired comparisons or 2-way ANOVA for ABRs. I do not pretend to be an expert, but many graphs in the study compare 3 or 4 conditions, for example "no noise" with 1h post noise and 1-week post-noise (e.g Figure 2J, Figure 4E-G and others). I wonder if one-way ANOVA with multiple comparisons would be more appropriate and stringent in these cases. For ABR graphs, it would be preferable to have multiple p values that indicate significance by frequency, as in some cases significance must differ (Figure 6L; Figure 7). This can be done with a multiple comparison post-hoc test like Sidak.

In cases where more than two groups were being compared, we reanalyzed the data using one-way ANOVA. Updated p-values are now indicated in the text and figure legends.

For the ABR data, in many cases, the p-value for 2-way ANOVA is significant, but most or all of the post-hoc multiple comparisons are not. For the post-hoc comparisons that are significantly different, asterisks were added above the individual frequencies.

There is no supplementary material in the current manuscript, and it is likely that data generated already might be useful to bolster the conclusion of the study and make it more transparent. That includes negative results when obtained, or validation and additional characterization of the mouse strains generated in this study, especially the powerful Xirp2 DelCter, for example.

We prefer to display all of our data within the manuscript if possible, rather than in the supplement.

Methods include a de-noising process, but its goal is not stated. Was it used to visualize gaps? In general, a section about image acquisition and processing might be useful. How were samples mounted and imaged to quantify gaps? Were confocal stacks taken? Are images projections from stacks?

The goal of the de-noising process was to improve the quality of the images and to remove artifacts due to small movements of the air table on which the microscope sits. Improved image quality allows for easier visualization of gaps and enrichment of immunostaining in representative images. More details about image acquisition were added to the Methods section and figure legends now indicate whether images were obtained from individual optical sections or from a confocal stack.

4) The hearing results would benefit from a more detailed discussion. Compared to C57BL/6J, the CBA background does not show threshold shifts at 11-12 week, and shows much milder shifts at 20 and 40 week (Figure 7). This suggests that early hearing defects in the full KO are largely caused by a modifier locus in C57BL/6J, and also suggests that threshold shifts in the DelCter strain at 20 month ( Figure 6L) are unlikely to be caused by defective short XIRP2 protein alone.

These results were described in more detail in the Discussion.

Other comments:Figure 2A, B: the phalloidin and XIRPs panels must have been switched by mistake for the utricle.

This has been corrected.

Figure 1I: the tamoxifen caption looks wrong "2x <;:;Tamoxifen"?

The caption has now been corrected.

Figure H: please use same terminology for Β-actin-Egfp throughout the text and figures. I think the Egfp is C-terminal, so GFP-Actin in H is imprecise, and does not match panels K-L.

All mentions of GFP-Actin have been changed to Β-actin-EGFP.

Figure 4A, C: captions to indicate that these are Xirp2 mutants would be helpful. In E-F, indicating that plots refer to the cochlea would be useful too.

Captions have been added.

Figure 5E: what does γ-actin gap ratio mean? An explanation is missing in the legend.

An explanation has been added to the figure legend.

Figure 6G: please confirm that phalloidin and XIRP2 panels are not switched here as well, because phalloidin shows slight over-enrichment at tips, as XIRP2 in previous panels.

The panels are correctly labeled in this image. XIRP2 labeling is variable between cells and is occasionally enriched or absent at stereocilia tips.

Line 99: "To address the possibility that the decreased percentage of IHCs with gaps was not due to the death of..". Change for "was due to the..".

Changed as suggested.

Line 111: please cite the Narayanan reference when the Actb-Egfp strain is first introduced.

The citation was added.

Line 137: I do not think there is a mention of stereocilia gaps in Elmod1 mutants in Krey 2018 although other defects are detailed? This could be clarified. How did this strain become an "actin gap" model? Ptprq1 actin gaps had been reported previously by contrast.

The Krey 2018 Elmod1 paper did not describe stereocilia gaps in their mutant mice. Our paper is the first to report gaps in Elmod1 mutants.

Line 138: the context of human samples would be valuable to add, if possible. Are these aged, or noise-damaged samples?

These are samples from de-identified patients. We have no further information on these samples. For the purpose of demonstrating in principle that XIRP2 is also preset in human hair cells, be believe this level of information in sufficient.

Line 221: hearing loss from ABR threshold shifts in DelC strain. Please indicate stage and strain (C57BL/6J) in the legend at least, or better, in a caption as in Figure 7.

The stage and strain were added to the figure legend and caption.

Line 246: A 105dB for 30min regime and the context makes me wonder whether the text should read ".… background to TTS-inducing noise", and not " to PTS-inducing noise"? Did wild-type controls fully recover in this experiment?

The wild-type mice did not fully recover, and we believe “PTS-inducing noise” is the appropriate term.

Line 268: remove one occurrence of "contribution".

Changed as suggested.

Line 293: "Combined with the decrease in gaps we found following noise exposure…" Technically, there is an increase in gaps following noise exposure, so a more precise formulation would be "…we found after recovery from noise exposure".

Changed as suggested.

Line 298: besides possibly explaining the low frequency of Actb-Egfp induction, recycling of actin monomers may principally explain the surprising low intensity of Egfp signal at "repaired" gaps, at least compared to stereocilia tips.

Yes, this is a good point. This is now mentioned in the discussion.

Line 607: remove double period.

Changed as suggested.

Reviewer #3 (Recommendations for the authors):One key feature of the proposed mechanism is that monomeric γ actin is recruited to breaks by XIRP2. I have three concerns.– The first is that there is very little evidence that XIRP2 can bind to G-actin. Although actin immunoprecipitates with GFP-XIRP2, the association could be indirect. In addition, washing with 30 mM MgCl2 is not a common way of disassembling F-actin. Are there additional references using this approach or experimental evidence of effectiveness? Even so, it would still be possible that F-actin bound to XIRP2 is resistant to disassembly. A better approach would be to show direct binding of purified XIRP2 to G-actin using an in vitro binding assay.

We agree that the association between XIRP2 and actin could be indirect. The text has been updated to reflect this possibility.

With regard to the issue that washing with 30 mM MgCl2 is not a common way of depolymerizing actin: There are additional references in which washing with MgCl2 was used to depolymerize F-actin during co-immunoprecipitation, including one provides more direct evidence that washing with MgCl2 leads to F-actin depolymerization during co-immunoprecipitation. Example references are

https://www.ncbi.nlm.nih.gov/pmc/articles/PMC2188260/pdf/je165197.pdf

https://www.ncbi.nlm.nih.gov/pmc/articles/PMC2114783/pdf/jc1052833.pdf

However, we agreed with the reviewer that this is not the standard way of distinguishing between binding to filamentous and/or monomeric actin.

To better demonstrate that XIRP2 and its C-terminal domain interacts (directly or indirectly) with monomeric actin, we conducted a new set of experiments, in which we co-immunoprecipitated a polymerization-deficient γ-actin mutant (G13R) with XIRP2 (see updated Results).

– One strength of the noise damage approach is that breaks are clearly shown to be induced, so they don't just result from damage during dissection. The same cannot be said for breaks in utricular stereocilia, which are long and might easily be broken during dissection. Is it possible that XIRP2 mutant stereocilia are more fragile? In that case breaks lacking γ-actin may have arisen when tissue was dissected and fixed before the normal repair complex was recruited. The idea that XIRP2 recruits G-actin to breaks would be more compelling if this were also deficient in XIRP2 KO IHC stereocilia, particularly a week after noise damage when the number of breaks is increased from baseline.

We cannot exclude the possibility that XIRP2 mutant stereocilia are more fragile than those in WT mice, but to address that possibility that we were inducing breaks during the dissection process, all temporal bones were fixed before dissecting out organs of Corti and utricles, making it unlikely that the lack of γ-actin at gaps is due to lack of time for recruitment. Quantification of gaps in XIRP2 knockout utricles was also repeated in P20 samples to reflect this change in dissection process. Additionally, the reduction of γ-actin enrichment in gaps was also observed in gaps in inner hair cells following noise exposure (see updated results).

– The loss of γ-actin at breaks in mutant XIRP2 stereocilia could be indirect, stemming from changes to the structure of the stereocilia core (known to be disordered in the KO) and not the loss of XIRP2 at breaks. The δ-C-term mutant mouse is a nice approach that may well answer this question, but this mouse needs additional characterization before it is possible to interpret whether loss of γ-actin or mutant XIRP2 from breaks is due to loss of the LIM domains or just defects in the core. Characterization could include comparing stereocilia height, width, and possibly F-actin organization to WT and KO mice. The more critical experiment is measuring the number of breaks in XIRP2 δ-C-term IHC stereocilia before and after noise, and assessing γ-actin levels in those IHC stereocilia breaks.

We agree that further characterization of the XIRP2-∆Cterm strain would strengthen our study. Unfortunately, we have had difficulty in obtaining enough XIRP2-∆Cterm mice due to unexplained breeding issues including infrequent litters and small litter sizes. Therefore, we prioritized determining whether the baseline level of gaps is elevated in XIRP2-∆Cterm mice. As we would expect if XIRP2 is no longer able to be recruited to gaps to facilitate their repair, we found an elevated percentage of cells with gaps in both auditory and vestibular organs in these mice (see updated Results). Temporal bones were fixed before fine dissection of the organ of Corti and utricle to prevent the breakage of potentially weakened stereocilia during the dissection process.

The second part of the hypothesis is that XIRP2 is recruited to breaks via its LIM domains. The description of SFSS and LIM domains in the introduction was such an interesting comparison that it was disappointing not to see a direct test of whether XIRP2 LIM domains bind to stressed, strained, or damaged F-actin. Along those lines, it would be interesting if LIM domains from other proteins that do bind strained actin would detect stereocilia breaks when used as probes. Either finding would strengthen the paper considerably, though both are challenging experiments.

Please see response to reviewer #1 in this matter.

[Editors' note: further revisions were suggested prior to acceptance, as described below.]

The manuscript has been improved but there are some remaining minor issues that need to be addressed, as outlined below in the comments from Reviewer #1, before we can make a final decision:Reviewer #1 (Recommendations for the authors):In the revised manuscript, the authors conducted additional experiments to support their claims that XIRP2 (1) is mechanoresponsive like other LIM domain-containing proteins but uses a distinct non-LIM mediated localization mechanism, and (2) binds monomeric γ-actin. Strikingly, their laser ablation experiments in fibroblasts revealed that it is not the LIM domain of XIRP2 that is mechanoresponsive to laser-induced stress fiber strain sites, but rather the structurally uncharacterized C-terminal domain (CTD). Their new results support a mechanism by which XIRP2 is recruited to sites of damage within F-actin core through its CTD to mediate repair in inner hair cells (IHCs). The additional data substantially strengthen their paper and reveal new insights into the mechanism of XIRP2-mediated repair of stereocilia. Furthermore, the authors' identification of a mechanism for rapid recruitment to mechanically damaged actin structures that does not depend on LIM domains is a very exciting development that is sure to stimulate additional work in the field.We believe the authors' revisions have addressed our previous concerns and the paper is now suitable for publication in eLife, pending minor textual revision to address the points below:1. In line 431, the authors include vinculin as one of the proteins modified ("tuned") by mechanical forces on F-actin, citing Mei et al. (2020). In that paper, only ɑ-catenin was observed to display force-activated F-actin binding, while vinculin was found to be insensitive to forces along F-actin. Thus, the authors should not include vinculin in this sentence.

In line 431, I removed the word “vinculin”, as it is not mechanosensitive on its own, as correctly pointed out by reviewer #1.

2. Lines 426-451: In a generally balanced fashion, the authors write that future work is necessary to determine the exact mechanosensitive domain in XIRP2's C terminus and the structural features that mediate its force-dependent interactions with F-actin. Aligned with this, we note that to our knowledge neither the individual LIM domains of testin nor XIRP2's force-responsive domain have been shown to directly bind tensed F-actin, which would require in vitro experiments with purified proteins. Some of the language in the discussion implies that this is the case. While it is of course very reasonable to speculate that this is true, we suggest that the authors be careful with their wording to make clear that this is currently a (leading) hypothesis.2.

It is our leading hypothesis that the CTD of XIRP2 constitutes a novel class of mechanosensitive protein domain. Further in vitro studies with purified proteins are needed to demonstrate that the mechanosensitivity of XIRP2’s CTD is an intrinsic property, and not mediated by an interacting protein. The CTD consists of ~470 amino acids, spanning half the length of the short isoform of XIRP2, and lacks any predicted peptide domains. Future studies will determine the exact mechanosensitive domain, and the structural features that mediate force-dependent interaction with F-actin. The CTD also helps recruit additional repair factors including monomeric actin and the crosslinker espin to gaps, suggesting that XIRP2 functions not only as a damage sensor, but also as an organizer of the subsequent actin remodeling efforts.